# BETTER KNOW NOTHING THAN HALF-KNOW ANYTHING: A PRECISE AND EFFICIENT DATASET FOR SCIENTIFIC REASONING IN LANGUAGE MODELS

## ABSTRACT

Large Language Models (LLMs) have achieved remarkable progress in reasoning tasks, *i.e.*, coding and mathematics. However, their ability to perform scientific reasoning remains significantly limited, probably hampered by the scarcity of high-quality scientific reasoning datasets. Existing approaches either rely on LLM-generated synthetic data (suffering from noise and hallucinations) or human-compiled documents (facing scarcity and non-standardization). In this paper, we empirically verify that integrating precise knowledge from original scientific documents with formalized questions and consistent answers can mitigate the need for large-scale data. Based on this insight, we design *PreciSci*, a pipeline for constructing multi-disciplinary scientific reasoning datasets. This pipeline involves extracting knowledge from reliable sources, refining questions for completeness and precision, applying multi-stage filtering to eliminate redundancy and noise, and refining answers to ensure reliable supervision. Leveraging PreciSci, we build *Open-Sci*, a precise and knowledge-dense scientific reasoning dataset. Experimental evaluations show that despite Open-Sci being less than one-sixth the size of state-of-the-art scientific reasoning datasets, it enables LLMs to achieve approximately **4.49**% better performance across diverse discipline-specific benchmarks.

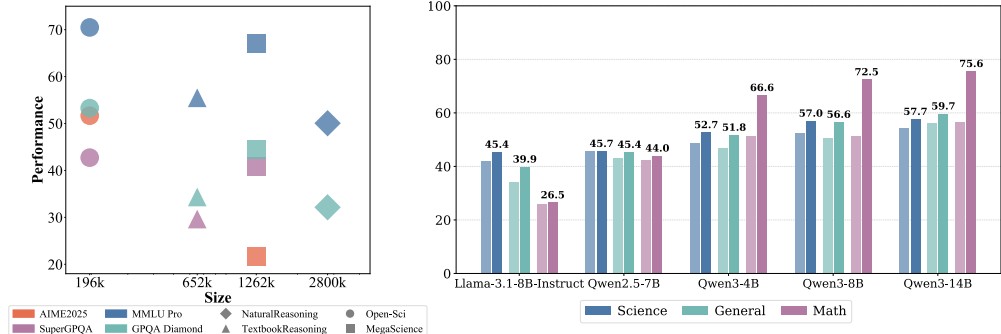

Figure 1: Left: Dataset size versus benchmark performance. Each point represents a dataset–benchmark pair, with colors for benchmarks and shapes for datasets. Despite its compact size (196k), Open-Sci achieves higher or comparable performance relative to much larger datasets. Right: Comparison of model performance across scientific, general, and mathematical benchmarks. Within each category, the lighter bars indicate the base model while the darker bars show the same model fine-tuned on our Open-Sci dataset. The consistent rightward shift across all groups demonstrates that Open-Sci fine-tuning yields performance improvements in every domain and across different models.

## 1 INTRODUCTION

The rapid advancement of Large Language Models (LLMs) (Xiang et al.; El-Kishky et al., 2025) has significantly enhanced their reasoning capabilities (Chen et al., 2025a), particularly in coding

and mathematical tasks (Shao et al., 2024; Ren et al., 2025). Equipped with long, well-constructed Chain-of-Thoughts (CoTs) (Wei et al., 2022) and Reinforcement Learning (RL) methods (Guo et al., 2025; Yu et al., 2025), LLMs can accurately solve complex algebraic equations, calculus problems, and generate functional code snippets or design software modules based on user prompts. However, their scientific reasoning abilities lag far behind (Wang et al., 2023; Xi et al., 2025), struggling to apply principles in physics, chemistry, or biology, which limits their use in scientific research. The primary underlying reason for these limitations lies in the lack of effective scientific reasoning datasets, whose construction is challenging because of difficulties in data verification, complex building processes, intricate data structures, and high risks of ambiguity and non-unique answers.

To address the scarcity of scientific reasoning data, prior efforts can be categorized into two classes: (1) Leveraging LLMs to generate synthetic data by either directly querying them for question-answer pairs (Wan et al., 2024) or providing reference papers for reasoning data generation (Chen et al., 2025b). Although scaling data size easily, these approaches face significant quality issues, including noisy, error-ridden content, susceptibility to model biases, and the risk of hallucinations. Given the stringent demand for precision in the scientific field, these flaws are particularly detrimental in scientific reasoning tasks, where minor inaccuracies can disrupt LLM training. (2) Extracting human-compiled question-answer pairs (Saikh et al., 2022; Singh et al., 2024) from scientific documents, *e.g.*, textbooks, examinations, and competitions. While ensuring better quality, this method faces data scarcity issues, as existing sources cover limited scenarios and often feature non-standardized questions with unclear requirements and incomplete reasoning in answers, potentially reducing the dataset to mere factual recall tasks.

To tackle the problems, we have empirically discovered that incorporating original and precise knowledge from scientific documents, along with refined questions and answers, can effectively reduce the demand for data scaling. As Friedrich Nietzsche once said, "Better know nothing than half-know many things." Acquiring accurate and comprehensive knowledge is much more important than accumulating a large amount of superficial information. The knowledge sourced from original scientific documents guarantees data precision and authenticity, while formalized questions and consistent answers structure the training data to foster effective learning and prevent it from devolving into mere factual recall tasks. By focusing on quality rather than quantity, we can build a small yet powerful scientific reasoning dataset that not only improves the performance of LLMs in scientific reasoning but also reduces the overhead of training resources.

In this paper, we carefully design a pipeline for multi-disciplinary scientific reasoning dataset construction, named *PreciSci*, which addresses the challenges of dataset construction and improves the quality and effectiveness of scientific reasoning datasets. Under this strategy, we first extract relevant knowledge from a wide range of authoritative scientific documents, including textbooks and examinations. Next, we conduct question formalization, where questions are categorized and clarified to ensure completeness and precision. We then enforce a rigorous multi-stage noise mitigation, which strictly eliminates redundancy, trivial samples, and potential contamination. Finally, we perform answer consistency to guarantee that answers remain precise and consistent while capturing the detailed reasoning process. By leveraging this pipeline, we filter out noisy or low-quality data and construct a precise, efficient, and knowledge-dense dataset, named *Open-Sci*. Despite having a dataset size that is less than $\frac{1}{6}$ of existing state-of-the-art (SOTA) datasets, *Open-Sci* achieves a performance that is approximately $4.49\%$ better than these SOTA datasets in various scientific reasoning evaluation tasks. This demonstrates the effectiveness of our pipeline and the superiority of focusing on data quality rather than quantity in improving the scientific reasoning capabilities of LLMs.

The main contributions of this paper are as follows:

- **Systematic data collection and processing pipeline.** We design a multi-stage pipeline motivated by the need for precision in scientific reasoning. The pipeline gathers data from textbooks, examinations and competitions by AI-human collaborative process. We formalize the questions and further processed by rigorous noise mitigation and answer consistency. This ensures both reliability and compatibility for scientific LLM training.

- **Compact and balanced scientific dataset.** Based on the pipeline, we construct *Open-Sci*, a 196k-instance dataset covering four natural sciences and 47 sub-disciplines. Each instance

is annotated with discipline and difficulty labels, yielding a precise and knowledge-dense corpus that provides high per-sample efficiency.

- **Improved performance and open contributions.** Fine-tuning on *Open-Sci* achieves stronger results with far less data: despite having only 196k samples ($\sim$ **16**% of Mega-Science), it surpasses the MegaScience baseline by **4.49**% on scientific benchmarks and yields striking gains on general and math tasks (e.g., AIME2025 +**30.00**%). We will release the dataset, trained models, pipeline and evaluation configurations to support reproducibility and further progress in open scientific AI.

## 2 RELATED WORK

**Scientific Datasets.** Parallel to model development, scientific datasets have evolved from narrow biomedical resources to large-scale, multi-disciplinary benchmarks emphasizing reasoning. Early datasets such as BioASQ (Nentidis et al., 2022) and PubMedQA (Jin et al., 2019) provide high-quality biomedical question-answering benchmarks but lacked cross-domain diversity. ScienceQA (Saikh et al., 2022) introduced chain-of-thought annotations to facilitate the training of reasoning processes, while SciBench (Wang et al., 2023), targeting university-level problems in physics, chemistry, and mathematics, systematically revealed the deficiencies of existing models in multi-step reasoning. More recent datasets like TextbookReasoning (Fan et al., 2025) and NaturalReasoning (Yuan et al., 2025) have respectively emphasized high-quality data sources and large-scale expansion. However, the former suffers from an imbalanced disciplinary distribution, while the latter faces issues with noise and inadequate difficulty control in its automated question generation process. In summary, scientific datasets have progressed from small-scale, single-domain resources to large-scale, multi-domain benchmarks that emphasize reasoning chains. Despite this progress, achieving both disciplinary balance and high-quality annotations remains a key challenge in advancing scientific models.

**Scientific Models.** Early large-scale scientific models often focused on a single discipline. For example, BioGPT (Luo et al., 2022) and PubMedGPT (Bolton et al., 2024) specialized in the biomedical domain, significantly improving performance on tasks like PubMedQA (Jin et al., 2019), but they lacked versatility and interdisciplinary capabilities. Subsequent efforts such as Galactica (Taylor et al., 2022) broadened the scope by pre-training on large-scale scientific corpora across medicine, chemistry, and mathematics, achieving state-of-the-art performance at the time but still falling short in terms of long-chain reasoning depth. More recently, scientific models have shifted towards balancing general reasoning ability with domain-specific expertise. Intern-S1 (Bai et al., 2025), by incorporating a large proportion of scientific text into a trillion-token scale corpus and combining it with large-scale reinforcement learning, has demonstrated performance comparable to closed-source models in disciplines such as physics, chemistry, biology, and medicine. After reinforcement reasoning fine-tuning, LLaMA-Nemotron (Bercovich et al., 2025) surpassed the best open-source models on scientific question-answering benchmarks like GPQA (Rein et al., 2024). Concurrently, new reward modeling frameworks like POLAR (Dou et al., 2025) have enhanced the stability and generalization of complex reasoning training. Overall, scientific models are evolving from single-discipline expert systems to foundational scientific models that possess both general and multidisciplinary capabilities.

## 3 PRECISCI

Scientific reasoning requires precise and high-quality data. Large but noisy corpora often introduce ambiguity and inconsistency, which limit the development of reliable reasoning ability. To overcome this, we design PreciSci, a data curation pipeline built for precision and efficiency rather than scale. The overview of our pipeline is presented in Figure 2. More details is in Appendix D

### 3.1 AUTHORITATIVE SOURCES

To construct a dataset emphasizing scientific precision over sheer scale, our pipeline begins with authoritative sources, including textbooks, standardized exams, research exercises, and subject-specific competitions. Textbooks provide systematically organized, expert-authored knowledge; exams and exercises supply well-structured problems that test conceptual accuracy; and competitions introduce advanced challenges requiring creativity and rigor. Drawing on these sources ensures

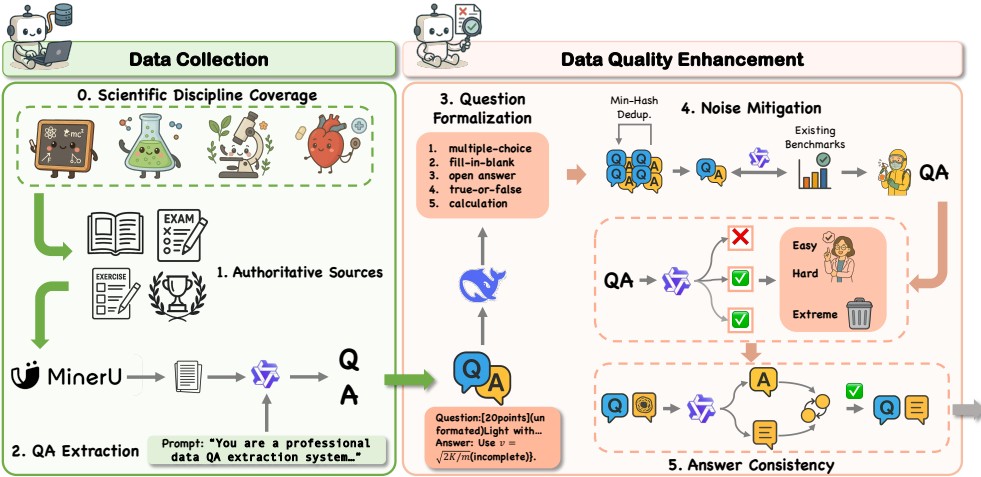

Figure 2: Overview of the PreciSci pipeline for constructing the Open-Sci dataset. It proceeds through five stages: collecting data from authoritative sources, extracting question–answer pairs with a hybrid AI–human approach, standardizing questions by types, mitigating noise through deduplication, decontamination, and filtering, and enforcing answer consistency via distillation. The result is a compact, scientifically rigorous dataset for advanced reasoning.

collected data is scientifically reliable, accurate, and inherently high-quality, laying a solid basis for further curation. For PDF-only materials, we employed MinerU (Wang et al., 2024a). For raw text, we directly incorporated it into the pipeline. This dual strategy ensured high recall across heterogeneous data formats. To extract question–answer (QA) pairs, we used Qwen2.5-72B (Qwen et al., 2024) with carefully designed extraction prompts to identify QA pairs, followed by human verification. This approach differs fundamentally from the reference-driven QA generation strategies, where models are asked to generate new questions given reference documents. Our method extracts existing scientifically valid problems that preserve their original rigor. This ensures that the pipeline benefits from the strengths of large models in pattern recognition while avoiding the risks of hallucination or uncontrolled reformulation (Ji et al., 2023). This hybrid AI-human process further reinforced the precision of data: the outcome is a curated collection of QA data that faithfully reflects authoritative sources and provides a solid basis for the subsequent stages of formalization and consistency.

## 3.2 QUESTION FORMALIZATION

Raw problems, even from authoritative sources, may still suffer from incomplete context or ambiguous phrasing. To ensure precise data representation, we avoid generating new questions and instead formalize the originals to preserve their meaning while improving clarity. Questions are categorized into five types, *i.e.*, multiple-choice, fill-in-the-blank, short-answer, true-or-false, and calculation, using Qwen2.5-72B. Then we formalize questions through type-specific prompts with DeepSeek V3-235B (Liu et al., 2024): clarifying options in multiple-choice, ensuring complete context in fill-in-the-blank, and enforcing numerical accuracy in calculation. For short-answer questions, we added minimal clarifications to guarantee that the prompt provides sufficient context for a unique and scientifically valid response. For true-or-false questions, we ensured that each statement is expressed in a precise and unambiguous form, avoiding vague qualifiers and enforcing scientific correctness. This formalization produced uniformly formatted, semantically unambiguous, and scientifically precise questions, strengthening the dataset's reliability for advanced reasoning.

## 3.3 NOISE MITIGATION

Although collected from authoritative sources, the raw collection still contained substantial redundancy and problematic items. On the one hand, many questions were near-duplicates that differed only in superficial phrasing, which inflate dataset size without adding new supervision value. On the other hand, some problems were overly trivial, providing little challenge for training, or flawed in

Table 1: Dataset retention after Noise Mitigation (deduplication, decontamination, and cascaded filtering). The pipeline reduces $594, 930$ raw Q–A pairs to $195, 681$ high-quality instances ($32.89\%$), as the filtering removes redundancy, triviality, contamination and ill-posed items.

| Stage | Physics | Chemistry | Biology | Medicine | Total | Retention Rate (%) |
|-------|---------|-----------|---------|----------|-------|--------------------|
| Raw Q - A Pairs | 144,844 | 113,018 | 154,147 | 182,921 | 594,930 | 100.0 |
| +Deduplication | 124,566 | 100,586 | 110,986 | 139,020 | 475,158 | 79.87 |
| +Decontamination | 92,499 | 85,603 | 85,793 | 92,743 | 356,638 | 59.95 |
| +Difficulty Filtering | 44,601 | 54,554 | 45,885 | 50,641 | 195,681 | 32.89 |

Table 2: Comparison of representative scientific reasoning datasets. D denotes digital-based sources (e.g., websites, existing datasets and digital documents); T denotes textbooks; E denotes exams or competitions. "Disc. Ann." indicates whether the dataset provides discipline-level annotations (✓ = available, ✗ = not available, Partial = partially available). "Ans. Len." is the average length of answers in tokens. "Open Scope" denotes the extent of open-source release (☆ = Data only, ★ = Data + Model, ◆ = Data + Model + Pipeline). The "#Discipline" column shows the number of disciplines, and for Open-Sci, the value "(47)" indicates four disciplines with a total of 47 sub-disciplines.

| Dataset | Source | Num | Non-math Cov. | Ans. Len. | #Discipline | Disc. Ann. | Open Scope |
|---------|--------|-----|---------------|-----------|-------------|-----------|------------|
| WebInstruct (Yue et al., 2024) | D | 13M | 32% | 266.43 | 8 | ✗ | ★ |
| NaturalReasoning (Yuan et al., 2025) | D | 2.8M | < 55% | 766.33 | 16 | ✗ | ☆ |
| Nemotron-Science (Bercovich et al., 2025) | D | 708.9k | 100% | 1716.50 | 1 | ✗ | ★ |
| TextbookReasoning (Fan et al., 2025) | T | 652K | 35% | 409.66 | 7 | ✓ | ◆ |
| MegaScience (Fan et al., 2025) | D,T | 1.25M | - | 692.93 | 7 | Partial | ◆ |
| **Open-Sci (ours)** | D,T,E | 196k | 100(%) | 1123.92 | 4(47) | ✓ | ◆ |

design, such as missing key conditions or being ambiguously stated. Both types of noise undermine effective learning by promoting shortcut memorization instead of genuine reasoning. To reduce such noise, we first applied locality-sensitive min-hashing (Mou et al., 2023) to remove near-duplicates within each discipline. To safeguard evaluation reliability, we performed strict benchmark decontamination by retrieving nearest neighbors via embedding similarity and confirming semantic overlap with LLM (Fan et al., 2025; Yuan et al., 2025), discarding all overlapping items. Finally, we further enforced scientific precision through cascaded filtering: a smaller model eliminated trivial questions based on empirical accuracy thresholds, while a stronger model re-examined the remainder to discard ill-posed or unreliable items. As shown in Table 1, these rigorous procedures retained only 32.89% of the original collection, highlighting the selectivity of our pipeline and ensuring that the resulting corpus consists of clean and precise data for reliable training.

## 3.4 ANSWER CONSISTENCY

Directly adopting raw reference answers can lead to noisy supervision: many raw answers are verbose, inconsistently formatted, or loosely aligned with their questions, undermining scientific precision. To address this, we introduced an answer consistency pipeline based on distillation. Concise final answers were first extracted from raw references, after which Qwen3-30B-A3B-Instruct (Yang et al., 2025) was used to generate and evaluate candidate responses paired with each question. Only correct candidates consistent with the references were retained. This consistency not only removes imprecise or noisy answers but also enforces clarity, alignment, and uniformity, while ensuring that the retained answers provide detailed reasoning, thereby enhancing the scientific reliability of model training.

## 3.5 DATASET DESCRIPTION

Using our PreciSci pipeline, we constructed Open-Sci, it contains 195,681 scientifically precise question–answer pairs spanning four domains (Physics, Biology, Chemistry, and Medicine) and 47 sub-disciplines. As summarized in Table 2, Open-Sci provides complete non-mathematics coverage with discipline-level annotations. This focus is advantageous because it avoids diluting training with generic mathematical problems and instead delivers data that is both precise and diverse within the core natural sciences. As illustrated in Figure 4, Open-Sci covers a broad spectrum of sub-disciplines,

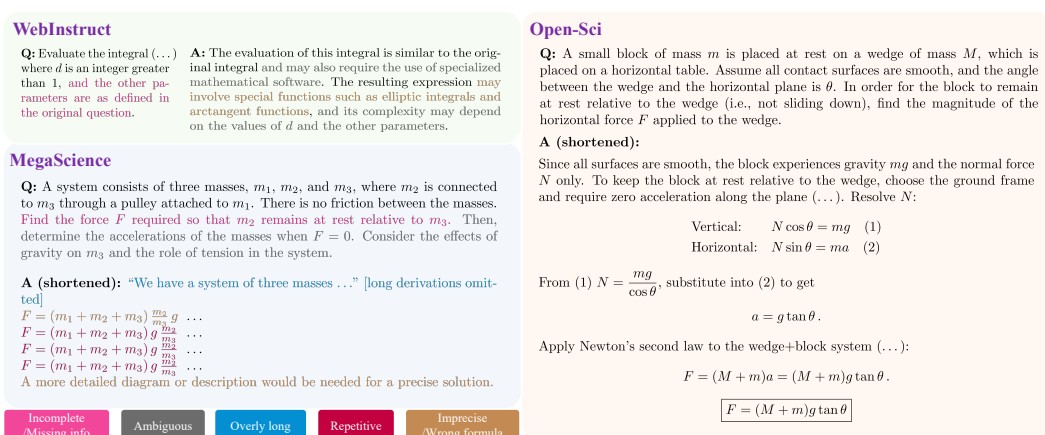

Figure 3: Quality comparison of QA pairs across WebInstruct, MegaScience, and Open-Sci.

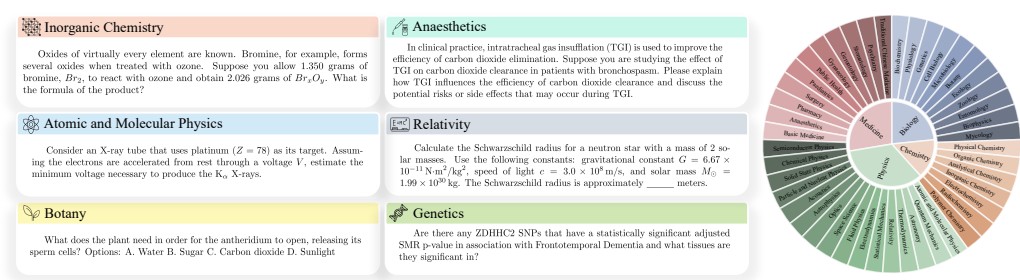

(a) Diverse question in Open-Sci    (b) Diverse sub-discipline in Open-Sci

Figure 4: (a) shows representative question examples across various sub-disciplines, illustrating the heterogeneity of scientific problem types. (b) depicts the disciplinary and sub-disciplinary coverage of the dataset, highlighting its broad distribution across physics, chemistry, biology, and medicine.

thereby offering rich and heterogeneous signals tailored for scientific reasoning. And it delivers answers that are substantially longer and more detailed than most prior datasets, while remaining scientifically accurate and aligned. Compared with previous scientific reasoning datasets, Open-Sci demonstrates clear qualitative advantages (Figure 3). While datasets such as WebInstruct (Yue et al., 2024) and MegaScience (Fan et al., 2025) frequently exhibit issues such as incomplete context, ambiguous formulations, overly long or repetitive answers, and imprecise formulas, Open-Sci is less prone to these problems. This case study highlights how careful curation and precise formalization result in higher-quality supervision, reducing noise and enabling more reliable learning. Overall, Open-Sci serves as a compact yet reliable dataset that prioritizes precision over scale, offering high-efficiency training resources for scientific reasoning. Additional distributions, such as difficulty levels and question type categories, are provided in the Appendix A.

## 4 EXPERIMENT

### 4.1 EXPERIMENTAL SETUPS

**Benchmarks.** We systematically assess the effectiveness of our proposed Open-Sci by evaluating trained models on four discipline-specific benchmarks: MedQA (Jin et al., 2020) and PubMedQA (Jin et al., 2019) for medicine, PHYSICS (Feng et al., 2025) for physics, ChemBench (Mirza et al., 2024) for chemistry, and ProteinLMBench (Shen et al., 2024) for biology. To broaden coverage beyond core disciplines, results on SuperGPQA (Du et al., 2025) and MMLU-Pro (Wang et al., 2024b) are further reported at the sub-discipline level. In addition, to examine generalization, we also include GPQA-Diamond (Rein et al., 2024), AIME2025 (American Institute of Mathematics, 2025), and Math-500 (Hendrycks et al., 2021), which measure general reasoning and mathematical ability.

Table 3: Performance comparison across scientific benchmarks (Medicine, Physics, Chemistry, Biology). Best scores are in **bold**, second best are underlined.

| Discipline | Benchmark | Qwen3-4B-Instruct | Qwen3-4B-Meg.Sci. | Qwen2.5-7B-Instruct | Qwen3-8B-Instruct | Qwen3-8B-Meg.Sci. | Qwen3-4B-Open-Sci | Qwen3-8B-Open-Sci |
|---|---|---|---|---|---|---|---|---|
| Medicine | MedQA | 68.30 | 70.12 | 67.68 | 75.33 | 77.32 | 71.82 | **77.43** |
| | PubMedQA | 72.60 | 76.30 | 75.10 | 75.50 | **77.90** | 77.00 | 77.60 |
| | SuperGPQA-Med. | 30.17 | 31.11 | 31.43 | 35.67 | **39.97** | 33.70 | 37.09 |
| Physics | PHYSICS | 23.20 | 21.07 | 14.52 | 25.17 | 25.02 | 29.97 | **36.75** |
| | MMLU-pro-Phy. | 67.51 | 69.28 | 57.74 | 73.06 | 75.44 | 78.37 | **82.37** |
| | SuperGPQA-Phy. | 30.79 | 31.67 | 24.11 | 36.13 | 36.98 | 41.51 | **46.36** |
| Chemistry | ChemBench | 47.68 | 50.30 | 43.39 | 52.36 | 54.18 | 52.13 | **55.28** |
| | MMLU-pro-Chem. | 68.64 | 70.67 | 54.15 | 72.26 | 75.18 | 78.36 | **82.51** |
| | SuperGPQA-Chem. | 26.40 | 27.87 | 23.29 | 30.02 | 33.35 | 36.69 | **42.17** |
| Biology | ProteinLMBench | 55.72 | 55.51 | 57.10 | 57.63 | 56.67 | 56.99 | **58.69** |
| | MMLU-pro-Bio. | 76.29 | 78.24 | 70.99 | 82.57 | 81.59 | 82.01 | **83.54** |
| | SuperGPQA-Bio. | 32.32 | 34.82 | 26.43 | 35.71 | 37.95 | 34.82 | **40.89** |
| Avg | | 47.91 | 51.41 | 42.80 | 52.36 | 53.97 | 54.22 | **58.46** |

**Compared Models and Scientific Reasoning Datasets.** We illustrate the effectiveness of Open-Sci by training it on current state-of-the-art LLMs, including Qwen2.5 (Qwen et al., 2025), Qwen3 (Yang et al., 2025), LLama-3.1 (Grattafiori et al., 2024) series models. We also compare Open-Sci with other scientific reasoning datasets, *i.e.*, NaturalReasoning (Yuan et al., 2025), MegaScience (Fan et al., 2025), and WebInstruct (Yue et al., 2024).

**Training setup.** We fine-tune Llama3.1-8B-Instruct (Grattafiori et al., 2024), Qwen2.5-7B-Instruct (Qwen et al., 2025) and Qwen3-4B, 8B, 14B (Yang et al., 2025) on the Open-Sci dataset. Training is implemented with MS-Swift (Zhao et al., 2025). Unless otherwise specified, we use a learning rate of $1 \times 10^{-5}$, set the global batch size to 128, and train models with 2 epochs using a warmup ratio of 0.05. All experiments are conducted on NVIDIA A100 GPUs.

**Evaluation setup.** Inference is carried out using LMDeploy (Contributors, 2023a), and all evaluations are performed with the OpenCompass (Contributors, 2023b) framework. For tasks requiring LLM-as-a-judge, we employ Qwen2.5-72B-Instruct (Qwen et al., 2025) to ensure consistent and reliable scoring. Due to the small sample sizes of AIME2025, GPQA-Diamond, and Math-500, we adopt a repeated evaluation. We report average@4 results for AIME2025 and GPQA-Diamond, and average@2 results for Math-500. All evaluations are conducted under a zero-shot setting. For decoding, Qwen2.5 and Llama use greedy decoding, while the Qwen3 family follows the official recommended setting with the temperature fixed at 0.7.

## 4.2 MAIN EVALUATION RESULTS

**Consistent improvements on scientific reasoning benchmarks.** Fine-tuning on Open-Sci consistently improves performance across all scientific benchmarks. As reported in Table 3, Qwen3-8B-Open-Sci improves PHYSICS from 25.17% to **36.75%**, MMLU-Pro-Chemistry from 72.26% to **82.51%**, with comparable gains in Biology and Medicine. Specifically, on SuperGPQA-Biology, the performance improves from 35.71% to **40.89%**, and on MedQA, it increases from 75.33% to **77.43%**. Even the smaller Qwen3-4B benefits from Open-Sci (e.g., PHYSICS +**6.77%**, ChemBench +**4.45%**). These consistent advances across all four domains demonstrate that the systematically curated organization of Open-Sci provides a reliable and effective foundation for model training. Importantly, despite its broad disciplinary coverage, Open-Sci does not dilute performance in any individual domain; instead, it enables comprehensive improvements across physics, chemistry, biology, and medicine. This shows that Open-Sci supplies not only precise and high-quality data but also balanced and well-structured scientific coverage, making it a practically effective resource for advancing scientific reasoning.

**Boosting performance on general reasoning and mathematics.** Although Open-Sci is primarily designed for scientific reasoning, models fine-tuned on it perform well across broader domains. As illustrated in Table 4, on MMLU-Pro and SuperGPQA, Open-Sci-8B-Instruct achieves strong results (**70.48%** and **42.73%**, respectively), and leads on GPQA-Diamond. The most notable

Table 4: Performance comparison on General and Math benchmarks. Style: **best**, second best.

| Model | General | | | Math | |
|---|---|---|---|---|---|
| | MMLU_Pro | SuperGPQA | GPQA_Diamond | AIME2025 | Math-500 |
| Qwen3-4B-Instruct | 59.78 | 34.05 | 45.96 | 18.33 | 84.70 |
| Qwen3-4B-MegaSci | 61.55 | 34.15 | 43.31 | 17.5 | 84.30 |
| Qwen2.5-7B-Instruct | 55.51 | 31.09 | 37.50 | 7.50 | 77.30 |
| Qwen3-8B-Instruct | 65.52 | 38.51 | 47.73 | 16.67 | 86.10 |
| Qwen3-8B-MegaSci | 67.08 | 40.81 | 44.57 | 21.67 | 87.00 |
| Qwen3-4B-Open-Sci | 65.96 | 37.57 | 49.87 | 40.00 | 93.20 |
| Qwen3-8B-Open-Sci | **70.48** | **42.73** | **53.28** | **51.67** | **93.30** |

improvement is in mathematics: Open-Sci-8B-Instruct scores **51.67**% on AIME2025, more than double the MegaScience-trained model. We believe these cross-domain benefits arise because precise scientific problems require systematic reasoning steps, such as careful use of definitions, symbolic manipulation, and quantitative calculation, that are also fundamental to general and mathematical tasks. By training on scientifically rigorous and well-structured data, models learn transferable reasoning patterns that extend beyond individual domains. In this sense, the precise and organized nature of Open-Sci not only strengthens domain-specific scientific reasoning, but also provides a foundation that supports generalization to mathematics and broader reasoning challenges.

**Less precise data is more worthy than more noisy data.** As shown in Table 3 and Table 4, compared with MegaScience's $\sim 1.25$M mixed-source corpus, Open-Sci uses only 196k samples yet surpasses it on most benchmarks. On average across scientific benchmarks, Open-Sci surpasses the MegaScience baseline by more than $+\textbf{4.49}$%, despite using nearly about $\frac{1}{6}$ the size of MegaScience. Notably, substantial gains also appear on general and mathematical tasks, *i.e.*, GPQA-Diamond ($+\textbf{8.71}$%) and AIME2025 ($+\textbf{30.00}$%). We suggest that these advantages stem from Open-Sci's emphasis on precise, systematically curated data. As illustrated in Figure 3, other datasets like MegaScience and WebInstruct often suffer from incomplete or missing context, ambiguous formulations, overly long or repetitive answers, and imprecise formulas. Such noise reduces training data usefulness, limiting models' ability to develop robust reasoning skills. By removing these issues, Open-Sci offers more precise, scientifically rigorous data, enabling more reliable and efficient learning. This observation highlights how a much smaller corpus can still outperform substantially larger but noisier datasets.

**Border effectiveness on diverse models.** To further validate the border effectiveness of Open-Sci, we fine-tune several leading open-source models, including Llama and Qwen series. As shown in Table 5, we draw two main observations. First, Open-Sci fine-tuning consistently improves performance across different model families and parameter scales. For instance, Qwen3-8B achieves an average gain of $+\textbf{8.69}$%, with striking improvements on mathematics tasks, *e.g.*, AIME2025 ($+\textbf{35.00}$%). Qwen3-14B exhibits similarly large benefits, $+\textbf{7.16}$% on average. We also observe the improvement in Llama-3.1-8B ($+\textbf{3.11}$%) and Qwen2.5-7B ($+\textbf{0.68}$%). This stems from the dataset's combination of precise scientific formulation and balanced disciplinary coverage, allowing models of varying architectures and scales to consistently acquire more rigorous reasoning patterns. Second, improvement extent depends on the model's baseline reasoning capacity. The Qwen3 series, with stronger reasoning priors, achieves the most substantial gains, while Llama-3.1-8B and Qwen2.5-7B show relatively smaller improvements. We suggest that this is because scientifically precise data poses more complex reasoning challenges, which weaker models may struggle to fully exploit.

### 4.3 ABLATION STUDY

**Data Efficiency.** To further examine the data efficiency of Open-Sci, we conduct a comparison by sampling 200k instances from WebInstruct, NaturalReasoning, and MegaScience, matching the scale of our dataset. As shown in Table 6, models fine-tuned on Open-Sci consistently achieve the highest average performance across scientific, general, and mathematical benchmarks under the same data budget. This demonstrates that precise and carefully curated data provides substantially greater per-sample efficiency, clearly surpassing larger-scale but noisier alternatives.

Table 5: Generalization across **Science**, **General**, and **Math** benchmarks. Numbers are accurate (%). For rows fine-tuned on our dataset (-Open-Sci), we show per-task gain $\Delta$ (Open-Sci – Baseline) in parentheses and report the mean gain **Avg.** $\Delta$.

| Model | Science | | | | General | | Math | | Avg. $\Delta$ |
|---|---|---|---|---|---|---|---|---|---|
| | MedQA | PHYSICS | ChemBench | ProteinLMB. | MMLU-Pro | SuperGPQA | AIME2025 | Math-500 | |
| Llama-3.1-8B-Instr. | 65.60 | 7.65 | 40.14 | 55.40 | 48.08 | 20.50 | 0.00 | 52.10 | |
| Llama-3.1-8B-Open-Sci | 67.99 (+2.39) | 15.13 (+7.48) | 40.94 (+0.80) | 57.52 (+2.12) | 54.91 (+6.83) | 24.86 (+4.36) | 0.83 (+0.83) | 52.20 (+0.10) | **+3.11** |
| Qwen2.5-7B-Instr. | 67.68 | 14.52 | 43.39 | 57.10 | 55.51 | 31.09 | 7.50 | 77.30 | |
| Qwen2.5-7B-Open-Sci | 71.35 (+3.67) | 14.91 (+0.39) | 39.34 (-4.05) | 57.20 (+0.10) | 59.22 (+3.71) | 31.49 (+0.40) | 10.00 (+2.50) | 78.00 (+0.70) | **+0.68** |
| Qwen3-4B-Instr. | 68.30 | 23.20 | 47.68 | 55.72 | 59.78 | 34.05 | 18.33 | 84.70 | |
| Qwen3-4B-Open-Sci | 71.82 (+3.52) | 29.97 (+6.77) | 52.13 (+4.45) | 56.99 (+1.27) | 65.96 (+6.18) | 37.57 (+3.52) | 40.00 (+21.67) | 93.20 (+8.50) | **+6.74** |
| Qwen3-8B-Instr. | 75.33 | 25.17 | 52.36 | 57.63 | 65.52 | 35.99 | 16.67 | 86.10 | |
| Qwen3-8B-Open-Sci | 77.43 (+2.10) | 36.75 (+11.58) | 55.28 (+2.92) | 58.69 (+1.06) | 70.48 (+4.96) | 42.73 (+6.74) | 51.67 (+35.00) | 93.30 (+7.20) | **+8.69** |
| Qwen3-14B-Instr. | 79.43 | 31.94 | 44.20 | 61.76 | 69.67 | 42.57 | 25.00 | 88.30 | |
| Qwen3-14B-Open-Sci | 81.88 (+2.45) | 42.38 (+10.44) | 45.47 (+1.27) | 60.91 (-0.85) | 73.54 (+3.87) | 45.89 (+3.32) | 55.83 (+30.83) | 95.30 (+7.00) | **+7.16** |

Table 6: Performance comparison across scientific, general, and mathematical benchmarks using 200k sampled data from different datasets. Best scores are in **bold**, second best are underlined.

| Dataset | Science Avg. | General Avg. | Math Avg. |
|---|---|---|---|
| **Ours** | **61.15** | **53.28** | **72.49** |
| MegaScience | 57.70 | 48.61 | 58.95 |
| NaturalReason | 51.60 | 40.66 | 39.05 |
| WebInstruct | 50.83 | 36.62 | 25.15 |

Table 7: Ablation study on different pipeline components. "w/o" denotes removing the corresponding component. Results are averaged across science, general, and mathematics benchmarks.

| Variant | Science Avg | General Avg | Math Avg |
|---|---|---|---|
| Full pipeline (Ours) | 61.16 | 51.89 | 70.69 |
| w/o Question Formal. | 57.87 | 47.22 | 66.22 |
| w/o Answer Consist. | 53.96 | 42.30 | 38.14 |
| w/o Noise-Mit. | 58.15 | 49.12 | 64.99 |

**Effect of formalization and consistency.** To verify the effectiveness of our pipeline, we conduct an ablation study by removing question formalization and answer consistency. The results show that eliminating either stage leads to clear performance degradation, as ambiguous questions or misaligned answers reduce the clarity of training data. The impact is especially severe when answer consistency is removed, since noisy or verbose references can directly misguide model learning. These findings confirm refinement is essential, with answer consistency playing a particularly critical role in ensuring reliable data quality.

**Effect of Noise Mitigation.** We also conduct an ablation by removing the noise mitigation stage of the pipeline. The results show a noticeable decline in performance, as trivial or low-quality instances (e.g., ill-posed or unreliable questions) remain in the dataset and reduce the effectiveness of training. This experiment confirms that filtering noise is essential for maintaining dataset precise and ensuring that retained samples support reliable model learning.

**Instruction-tuned models show more efficient reasoning.** To examine how instruction tuning (Shengyu et al., 2023) affects model reasoning, we compare the original Qwen3-8B with the same model fine-tuned on Open-Sci under reasoning mode. As shown in Table 8, the Open-Sci–tuned model generates fewer reasoning tokens on average (2918 vs. 3262) while achiev-

Table 8: Comparison of reasoning efficiency. We report average output length (in token-level) and benchmark accuracy under reasoning mode.

| Model | Tokens | Science Avg. | General Avg. |
|---|---|---|---|
| Qwen3-8B-Think | 3262 | 59.97 | 58.34 |
| Ours (8B)-Think | 2164 | 61.03 | 59.74 |

ing higher accuracy on average. The scientific average rises from $59.97\%$ to $60.49\%$, the general average from $58.34\%$ to $59.52\%$. This shows that instruction tuning not only strengthens task-following and reasoning ability but also makes the reasoning process more concise and efficient.

## 5 CONCLUSION

We presented Open-Sci, a compact yet precise dataset systematically curated through the PreciSci pipeline. Despite its modest scale of 196k instances, Open-Sci consistently outperforms much larger corpora across scientific, general, and mathematical benchmarks, demonstrating that precision is more valuable than raw size. Our results highlight the importance of precise and high-quality data in advancing reasoning capabilities, and we release Open-Sci together with models and pipeline to support future research in open scientific AI.

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

## A  DATASET DETAILS

### A.1  STATISTICAL PROPERTIES

The internal statistics of Open-Sci are presented in Table 9. The dataset contains 195,681 instances distributed across four domains in a balanced manner: Physics 44,601 samples (22.79%), Biology 45,885 (23.45%), Chemistry 54,554 (27.88%), and Medicine 50,641 (25.88%). These domains are further divided into forty-seven sub-disciplines. The mean question length is 72.88 tokens and the mean answer length is 1,123.92 tokens. Physics exhibits the longest answers, averaging 1,565.56 tokens, while Biology is more concise with 830.54 tokens. These statistics confirm that Open-Sci maintains both domain-level balance and fine-grained sub-discipline diversity, providing heterogeneous and reasoning-rich supervision signals.

### A.2  DIFFICULTY DISTRIBUTION

Each instance in Open-Sci is annotated with a difficulty level derived from model-based rollouts during data curation, as described in Section 3.3. Figure 5 illustrates the final distribution across nine levels. The histogram shows that the dataset avoids being dominated by either trivial or excessively hard problems. Instead, it forms a graded spectrum that ranges from simple factual recall to complex multi-step reasoning. This balanced distribution ensures that Open-Sci provides training signals covering a wide range of complexity. Rather than concentrating on a narrow difficulty band, the dataset exposes models to a continuum of challenges, which is essential for developing robustness and generalization.

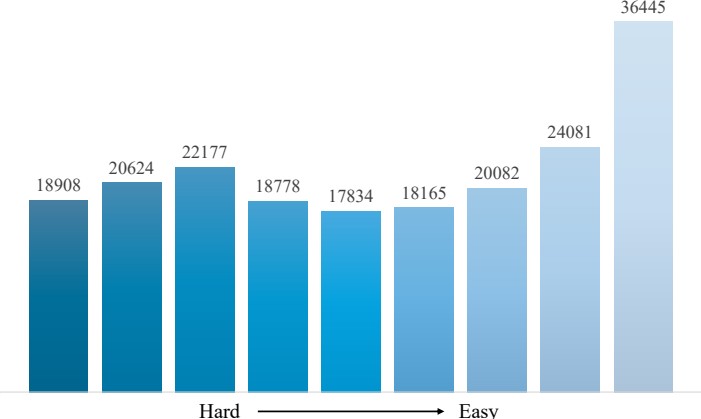

Figure 5: Distribution of difficulty.

### A.3  DIVERSITY

Open-Sci also exhibits substantial diversity in both question formats and answer styles. As illustrated in Figures 6a and 6b, the dataset covers a wide range of assessment types, including open-ended questions (48%), calculation problems (30%), multiple-choice items (13%), and smaller shares of true/false, fill-in-the-blank, and other formats. To characterize answer diversity, we categorize each reference answer by its token length: 1–5 tokens as Factoid, 6–20 as Phrase, 21–50 as Sentence, and over 50 as Paragraph. The resulting distribution contains 14% factoid answers, 23% phrases, 18% sentences, and 45% paragraphs. Shorter responses typically align with structured formats such

Table 9: Statistics of dataset across different domains. "Q. Len." is the average question length (in tokens); "Ans. Len." is the average answer length (in tokens); "#Sub-disc." denotes the number of sub-disciplines covered in each domain.

| Domain/Split | Samples | % of total | #Sub-disc. | Q. Len. | Ans. Len. |
|---|---|---|---|---|---|
| Physics | 44,601 | 22.79% | 16 | 89.34 | 1,565.56 |
| Biology | 45,885 | 23.45% | 11 | 54.51 | 830.54 |
| Chemistry | 54,554 | 27.88% | 7 | 80.35 | 1,145.30 |
| Medicine | 50,641 | 25.88% | 13 | 66.98 | 977.75 |
| Total | 195681 | 100% | 47 | 72.88 | 1,123.92 |

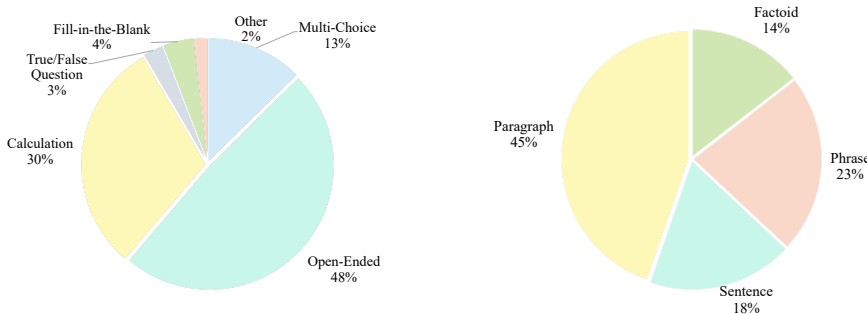

(a) Distribution of question types.  (b) Distribution of reference answer lengths.

Figure 6: Diversity of Open-Sci in terms of question types and reference answer lengths.

as multiple-choice or calculation, while longer ones arise from open-ended or analysis-style tasks requiring detailed explanations. This variety ensures that Open-Sci provides supervision signals for both concise factual queries and extended scientific argumentation, thereby promoting broader generalization across tasks.

## B    THE USE OF LARGE LANGUAGE MODELS (LLMS)

In this work, Large Language Models (LLMs) were used solely to polish the language for clarity and readability. No LLMs were employed for idea generation, experimental design, data analysis, or any other part of the research process.

## C    EVALUATION DETAILS

The complete evaluation on MMLU-Pro is in Table 10

## D    PIPELINE DETAILS

### D.1    QUESTION FORMALIZATION

The prompt for question formalization is in Figures 7

### D.2    NOISE MITIGATION

Duplicate or near-duplicate questions are removed using MinHash with locality-sensitive hashing. We adopt 3-gram shingles, a similarity threshold of 0.6, and LSH parameters of 1 band and 10 rows.

To avoid contamination with evaluation benchmarks, we encode all Open-Sci training samples and all benchmark items using the bge-large-en-v1.5 embedding model. For each training sample, we retrieve its top-5 most similar benchmark items under cosine similarity. Each retrieved pair (training item, benchmark items) is evaluated for semantic overlap using Llama-3.3-70B-Instruct with temperature

Table 10: Detailed results on MMLU-Pro.

| Benchmark | Qwen3-8B-Instr. | Qwen3-8B-Open-Sci |
|---|---|---|
| MMPU-Pro-Math | 83.94 | 90.67 |
| MMPU-Pro-Physics | 73.06 | 82.37 |
| MMPU-Pro-Chemistry | 72.26 | 82.51 |
| MMPU-Pro-Law | 34.51 | 38.24 |
| MMPU-Pro-Engineering | 52.32 | 62.33 |
| MMPU-Pro-Economics | 74.64 | 77.25 |
| MMPU-Pro-Health | 64.18 | 65.28 |
| MMPU-Pro-Psychology | 69.67 | 71.05 |
| MMPU-Pro-Business | 73.64 | 77.69 |
| MMPU-Pro-Biology | 82.57 | 83.54 |
| MMPU-Pro-Philosophy | 54.51 | 60.32 |
| MMPU-Pro-Computer Science | 69.51 | 75.12 |
| MMPU-Pro-History | 53.54 | 56.96 |
| MMPU-Pro-Other | 58.87 | 63.31 |

Table 11: Number of removed items per benchmark.

| Benchmark | ProteinLMBench | PHYSICS | MedQA | PubMedQA | ChemBench | MMLU-Pro | SuperGPQA | GPQA | Math | AIME-2025 |
|---|---|---|---|---|---|---|---|---|---|---|
| Nums | 3115 | 4708 | 16821 | 191 | 6683 | 32313 | 52594 | 1741 | 354 | 0 |

set to $0$ to ensure deterministic judgments. A training sample is removed if any retrieved benchmark neighbor is judged to exhibit semantic equivalence. The exact prompt used for semantic equivalence checking is provided in Figure 8. The numbers of removed items per benchmark are shown in Table 11

To further eliminate trivial, ambiguous, or ill-posed questions, we apply two-stage LLM-based filtering. Qwen2.5-7B-Instruct is used to remove overly easy items ($\geq 7$ out of $8$ sampled responses correct). Qwen3-30B-A3B-Instruct is used to filter scientifically invalid or ill-posed items (0 out of $8$ responses correct).

# E    CASES

More cases are shown in Figure 14.

# F    MORE ABLATION STUDY

## F.1    PERFORMANCE UNDER DIFFERENT COVERAGE.

We studied the effect of coverage on our Open-Sci: (i). Open-Sci-Med-50k: single-discipline. (ii). Open-Sci-Mixed-50k: 50k samples randomly from Open-Sci with full disciplines. It shows that expanding the coverage further improves the performance of the model.

Table 12: Performance under different disciplinary coverage.

| Benchmark | Qwen3-8B | Open-Sci-Med-50k | Open-Sci-Mixed-50k |
|---|---|---|---|
| ChemBench | 52.36 | 52.53 | **53.64** |
| ProteinLMBench | 57.63 | 58.37 | **58.79** |
| MedQA | 75.33 | **76.93** | 75.74 |
| PHYSICS | 25.17 | 29.91 | **33.23** |
| MMLU-Pro | 65.52 | 68.83 | **69.06** |

**Prompt for question formalization:**

### INPUT
Here is the markdown content that you need to analyze, correct, and filter:
Question: {question}
### GLOBAL TASKS
1. **Strip embedded answers**
-   If the correct answer or solution steps appear in the stem, remove them entirely.
2. **Fix formatting & LaTeX**
-   Scan all math expressions, symbols, superscripts/subscripts, fractions, etc., and correct any mis-parses (e.g.,
    "x2" → "x^2", missing backslashes).
3. **Normalize markdown**
-   Ensure headings, lists, code blocks and inline math are valid Markdown/LaTeX.
4. **Maintain Language**
-   Ensure the output question is in the same language as input_question

### PER-TYPE RULES
Current Question Type: {question_type}

#### 1. Multiple-Choice
-   Remove any duplicate option lists.
-   Ensure exactly one set of options labeled **A.**, **B.**, **C.**, **D.**, etc.
-   If labels are missing or inconsistent, insert or renumber them sequentially.
#### 2. True/False
-   If presented as a declarative statement, rephrase to:
> "Determine whether the following statement is true or false:
> _<original statement>_"
-   Remove any embedded "True" or "False" answers.
#### 3. Short-Answer
-   Confirm the question ends with a clear prompt (e.g. "What is…?", "Explain why…?").
-   Remove any parenthetical answer hints.
#### 4. Fill-in-the-Blank
-   Represent blanks consistently as `____` (at least 3 underscores).
-   Ensure each blank corresponds to exactly one missing answer.
-   Remove any answers shown inline.
-   If there are multiple blanks, number them:
> "The capital of France is (1) ____, and its currency is (2) ____."
#### 5. Calculation
-   Strip out worked-out solutions and final answers.
-   Verify units are present and correctly formatted (e.g., "m", "kg", "$").
-   Confirm numeric values in the stem are clear (e.g., no stray commas or spaces).
-   If a formula is required, ensure it's in LaTeX math delimiters.
### OUTPUT
Return exactly this, in Markdown, without surrounding triple-tick fences:
**Refined Question**
<your cleaned, corrected question markdown here directly without redudant information such as Question
Type>
**Rationale**
<Briefly note what you removed or fixed>

Figure 7: Prompt for question formalization.

## F.2   PERFORMANCE UNDER DIFFERENT TEACHER MODEL.

We further distilled the same data from Llama-3.1-70B-Instruct and fine-tuned Llama-3.1-8B. The results is shown in Table 13. In this setting, the Llama-distilled variant sometimes achieves slightly higher scores on ChemBench, but training directly on Open-Sci still yields the strongest results on most benchmarks. This pattern suggests that different teacher models emphasize different aspects of scientific reasoning, and that a hybrid or mixed-source distillation strategy may further improve Llama-family students.

**Prompt for decontamination**

I will now give you two questions: Original question and Candidate question. Please help me determine if the following two questions are the same.

Original question:
`<ORIGINAL_PROBLEM>`

Candidate question:
`<CANDIDATE_PROBLEM>`

Disregard the names and minor changes in word order that appear within.
If their question prompts are very similar and, without considering the solution process, they produce the same answer, we consider them to be the same question.

Output Format:
Analysis: [Provide a detailed analysis evaluating the similarity between these questions]
Decision: [YES/NO]

Figure 8: Prompt for decontamination.

Table 13: Evaluation results for Llama.

| Benchmark | Llama-3.1-8B | Llama-8B-Distill-70B | Llama-3.1-8B-Open-Sci |
|---|---|---|---|
| ChemBench | 40.14 | **50.41** | 40.94 |
| ProteinLMBench | 55.40 | 54.66 | **57.52** |
| MedQA | 65.60 | 66.61 | **67.99** |
| PHYSICS | 7.65 | 11.67 | **15.13** |
| MMLU-Pro | 48.08 | 52.56 | **54.91** |

**Prompt for judgement**

Please as a grading expert, judge whether the final answers given by the candidates below are consistent with the standard answers, that is, whether the candidates answered correctly.
Here are some evaluation criteria:
1. Please refer to the given standard answer. You don't need to re-generate the answer to the question because the standard answer has been given. You only need to judge whether the candidate's answer is consistent with the standard answer according to the form of the question. Don't try to answer the original question. You can assume that the standard answer is definitely correct.
2. Because the candidate's answer may be different from the standard answer in the form of expression, before making a judgment, please understand the question and the standard answer first, and then judge whether the candidate's answer is correct, but be careful not to try to answer the original question.
3. Some answers may contain multiple items, such as multiple-choice questions, multiple-select questions, fill-in-the-blank questions, etc. As long as the answer is the same as the standard answer, it is enough. For multiple-select questions and multiple-blank fill-in-the-blank questions, the candidate needs to answer all the corresponding options or blanks correctly to be considered correct.
4. Some answers may be expressed in different ways, such as some answers may be a mathematical expression, some answers may be a textual description, as long as the meaning expressed is the same. And some formulas are expressed in different ways, but they are equivalent and correct.
Please judge whether the following answers are consistent with the standard answer based on the above criteria. Grade the predicted answer of this new question as one of:
A: CORRECT
B: INCORRECT
Just return the letters "A" or "B", with no text around it.
Here is your task. Simply reply with either CORRECT, INCORRECT. Don't apologize or correct yourself if there was a mistake; we are just trying to grade the answer.
<Original Question Begin>:
{question}
<Original Question End>
<Gold Target Begin>:
{label}
<Gold Target End>
<Predicted Answer Begin>:
{prediction}
<Predicted End>
Judging the correctness of candidates' answers:

Figure 9: Prompt for judgement.

**Prompt for reference answer extraction(Biology):**

## Task Description
You are tasked with extracting the final reference answer from a detailed biological solution that contains both reasoning steps and the final conclusion.
Biological answers may take the form of:
- A single word/phrase (e.g., a term like chloroplasts, IL-2, mutation breeding).
- A letter corresponding to a multiple-choice option.
- A full-sentence or multi-sentence explanation (when the final answer includes description of a function, mechanism, or role).
Your goal is to output the final, standalone reference answer in its complete form, without losing essential biological context.
## Input Format
You will receive:
1. A biological question (Question)
2. A detailed answer that includes reasoning steps and the final conclusion (Detailed Answer)
## Output Requirements
- Extract only the final reference answer, without intermediate reasoning.
- Ensure the extracted answer is complete and self-contained (include the full descriptive conclusion if present, not just a fragment).
- If the answer contains both a choice label (e.g., "B.") and the explanatory phrase, extract both together (e.g., "B. Mutation breeding").
- If the answer is a descriptive explanation (e.g., about a cytokine's function), extract the entire explanatory block, not just the first sentence.
- Do not add new explanations or modify the original meaning.
- If multiple possible answers are given, select the one explicitly marked as final, preferred, or correct.
## Special Notes for Biology
- Many biological answers require context (e.g., naming a molecule and stating its function). Always keep the final explanatory part intact.
- For multiple-choice questions, do not drop the option letter if it is part of the final answer.
- For open-ended questions (e.g., mechanisms, processes), keep the full concluding explanation. Do not shorten to a single keyword unless the original final answer is only that keyword.
## Example 1(Multiple-choice)
### Question:
在线粒体中，主要通过哪种结构进行氧化磷酸化反应以生成ATP?
A. 基质
B. 外膜
C. 内膜
D. 膜间隙
### Detailed Answer:
…reasoning steps…
所以综合起来，答案应该是选项C，内膜。
### Reference Answer:
C. 内膜
## Example 2(Open-ended)
### Question:
在研究中发现CD2+细胞释放出一种因子能促进自身生长和延长存活期。这种因子最可能是什么？其主要功能和免疫作用是什么？
### Detailed Answer:
…reasoning steps…
1.这种因子最有可能是白细胞介素-2（Interleukin-2, IL-2）。
2.IL-2的主要功能及免疫作用：•促进T细胞增殖、分化和存活 •驱动效应和记忆T细胞形成 •调节免疫耐受并协同B细胞和NK细胞功能
### Reference Answer:
1.白细胞介素-2（Interleukin-2, IL-2)
2.功能及免疫作用：促进T细胞增殖、分化和存活；驱动效应与记忆T细胞形成；调节免疫耐受；协同B细胞和NK细胞功能。
## Instructions
1.Read the question carefully to understand what is being asked.
2.From the detailed answer, locate the final conclusion.
3.Extract the conclusion in full (including option labels or full descriptive text).
4.Output only the reference answer, clearly formatted.
## Question:
`<PROBLEM>`
## Detailed Answer:
`<ANSWER>`
Now process and return the result.

Figure 10: Prompt for reference answer extraction in biology.

**Prompt for reference answer extraction(Chemistry):**

## Task Description
You are tasked with extracting the final reference answer from a detailed chemistry solution that contains both reasoning steps and the final conclusion. Chemistry answers often differ from math answers, and the final reference answer may take one of several forms:
- A short explanatory statement (for conceptual/causal questions)
- True/False judgments (often listed part by part)
- Numerical results with correct units and significant figures (e.g., grams, mol, %, K values)
- Balanced chemical equations or ionic equations
- Standardized symbolic expressions (e.g., bold text, $\boxed{...}$, or clearly listed results)
## Input Format
You will receive:
1. A chemistry question (Question)
2. A detailed answer that includes reasoning steps and the final conclusion (Detailed Answer)
## Output Requirements
- Extract only the final reference answer, without reasoning steps.
- Ensure completeness: if the answer contains multiple sub-answers (e.g., parts a–d), extract all of them.
- Ensure stand-alone clarity: the extracted answer must be self-contained and usable as a standard solution.
- Preserve chemistry formatting:
   - Keep chemical equations as written (e.g., 2 NH3 ⇌ NH4+ + NH2-).
   - Keep numerical results with correct units and significant figures.
   - For multiple-choice or True/False questions, list answers clearly part by part.
- Remove redundancy: exclude repeated reasoning or irrelevant background.
- If explicit markers exist (e.g., "Final Answer", "Thus", $\boxed{}$), prioritize extracting from them.
- If no explicit marker exists, extract from the summary statement at the end.
## Example 1
### Question:
Aniline (conjugate acid pKa 4.63) is a considerably stronger base than diphenylamine (pKa 0.79). Account for these marked differences.
### Detailed Answer:
…reasoning steps…
### Final Answer: \boxed{\text{Aniline is a stronger base than diphenylamine due to less resonance delocalization of the lone pair on nitrogen.}}
### Reference Answer:
Aniline is a stronger base than diphenylamine due to less resonance delocalization of the lone pair on nitrogen.
## Example 2
### Question:
Label the following statements true or false: a. … b. … c. … d. …
### Detailed Answer:
…reasoning steps…
### Final Answer: a. \boxed{\text{False}}
b. \boxed{\text{False}}
c. \boxed{\text{True}}
d. \boxed{\text{True}}
### Reference Answer: a. False b. False c. True d. True
## Instructions
1.Read the question carefully to determine its type (numerical calculation / judgment / explanation / chemical equation).
2.Locate the final conclusion in the detailed answer (usually marked as "Final Answer" or at the end).
3.Extract the complete, self-contained final answer.
4.Format it clearly so it can be used directly as a reference answer.
## Question:
`<PROBLEM>`
## Detailed Answer:
`<ANSWER>`
Now process and return the result.

Figure 11: Prompt for reference answer extraction in chemistry.

**Prompt for reference answer extraction(Medicine):**

## Task Description
You are tasked with extracting the final reference answer from a detailed medical solution that contains both reasoning/discussion steps and the final conclusion. The reference answer should be precisely the definitive conclusion, phrased exactly as it would appear in a standard medical solution key.
## Input Format
You will receive:
1. A medical question
2. A detailed answer that includes reasoning steps and the final answer
## Output Requirements
- Extract ONLY the final reference answer, without reasoning or intermediate discussion.
- Ensure the reference answer is complete, clinically accurate, and able to stand alone.
- Keep the answer type consistent with the question:
   - For multiple choice questions: return the final selected option (e.g., "D. 受害者的血型")
   - For short-answer/fill-in-the-blank: return the concise term or phrase (e.g., "雌激素")
   - For essay/analysis: return the final summarized conclusion (e.g., "CNETs在形态、分子机制、临床进程及预后等方面均与实性pNETs存在显著差异，符合肿瘤分类中的独立类型标准。")
- Do not shorten the final answer if it requires full context to be medically correct (e.g., mechanisms, multi-part responses).
- Do not include reasoning steps, explanations, or "因此/所以/结论是"类提示语。
- If multiple answers are presented, extract the one explicitly marked as final, correct, or preferred.
## Extraction Priorities for Medical Content
1.Direct final statement (often at the end of the solution).
2.Option label if the answer is multiple choice (e.g., "A. M. tuberculosis").
3.Key medical term/phrase if fill-in-the-blank or short question.
4.Complete concluding paragraph if analytical/essay type.
## Example
### Question:
在法医病理学中，分析损伤模式对于确定死亡原因至关重要。当检查伤口时，法医病理学家需要考虑多种因素。下列哪一项不是考虑的因素？
A. 使用的凶器类型
B. 伤口的角度
C. 施力方向
D. 受害者的血型
### Detailed Answer:
... derivation ...
所以正确答案是D. 受害者的血型。
### Reference Answer:
D. 受害者的血型
## Instructions
1.Carefully read the question to determine the expected format (MCQ, short answer, or essay).
2.Analyze the detailed answer and locate the final conclusion or chosen option.
3.Extract only the final reference answer in its medically correct and complete form.
4.Return it cleanly formatted, with no extra commentary.
## Question:
`<PROBLEM>`
## Detailed Answer:
`<ANSWER>`
Now process and return the result.

Figure 12: Prompt for reference answer extraction in medicine.

**Prompt for reference answer extraction(Physics):**

## Task Description
You are tasked with extracting the final reference answer(s) from a detailed physics solution that contains both reasoning steps and final results. Physics solutions often include long derivations, intermediate formulas, and unit conversions. Your goal is to extract the **final, definitive answer(s)** that can serve as the standard solution.
## Input Format
You will receive:
1. A physics problem (the question)
2. A detailed solution (including reasoning, formulas, intermediate steps, and the final answer)
## Output Requirements
- Extract ONLY the final reference answer(s), not the reasoning or derivations.
- Ensure the reference answer includes:
  - The **numerical value(s) with correct unit(s)** when present.
  - The **final expression or conclusion** if the solution ends with a formula or statement.
  - **All sub-answers** if the problem is divided into multiple parts (e.g., (a), (b), (c)).
- Do NOT add explanations, derivations, or restatements of the problem.
- Do NOT omit essential parts of the final answer (e.g., units, subscripts, choice letters in multiple-choice).
- If multiple possible answers are given, choose the one explicitly marked as **final** (e.g., after "The final answer is:").
- If the answer is conceptual (True/False or qualitative), extract it exactly as written.
## Example 1 (Numerical)
### Question:
What is the area of a circle with radius 5 cm?
### Detailed Answer:
... derivation ...
Therefore, the area of the circle with radius 5 cm is 78.54 cm².
The final answer is: 78.54 cm²
### Reference Answer:
78.54 cm²
## Example 2 (Physics Formula)
### Question:
Write the secular equation for eigenfrequencies of small oscillations.
### Detailed Answer:
... derivation ...
The final answer is: $\det(K - \omega^2 M) = 0$
### Reference Answer:
$\det(K - \omega^2 M) = 0$
## Example 3 (Multiple parts)
### Question:
(a) Compute velocity … (b) State direction …
### Detailed Answer:
... derivation ...
The final answers are:  (a) 4.0 m/s²   (b) leftward
### Reference Answer:
(a) 4.0 m/s²  (b) leftward
## Instructions
1. Read the question carefully.
2. Identify the **explicit final answer(s)** in the detailed solution.
3. Extract them completely (including numbers, units, or words).
4. If multiple parts exist, present each part clearly and separately.
5. Do not copy any intermediate steps or reasoning—only the final results.
## Question:
`<PROBLEM>`
## Detailed Answer:
`<ANSWER>`
Now process and return the result.

Figure 13: Prompt for reference answer extraction in physics.

### Solid State Physics

Explain how the correlation strength in strongly correlated electron systems affects the quantum oscillations observed in magnetization and resistance under high magnetic fields.

### Astrophysics

A cataclysmic variable star system exhibits a periodic brightness variation during its superoutburst phase. This phenomenon, known as a superhump, is caused by the precession of an eccentric accretion disk. Given that the orbital period of the binary system is $P_{orb} = 0.15$ days and the superhump period is observed to be $P_{sh} = 0.153$ days, calculate the fractional change in the superhump period $\Delta P_{sh}/P_{sh}$. Furthermore, discuss how this fractional change can provide insights into the mass ratio $q = M_2/M_1$ between the secondary star ($M_2$) and the white dwarf primary ($M_1$).

### Organic Chemistry

An alkene which is least reactive towards electrophilic addition, among the following is (a) $H_2C = CH - Cl$ (b) $(CH_3)_2\, C = CH_2$ (c) $(CH_3)_2\, C = C\, (CH_3)_2$ (d) $ClCH_2CH = CH_2$

### Physical Chemistry

Assume you dissolve 0.303 g of benzoic acid in enough water to make 100 mL of solution and then titrate the solution with 0.178 M NaOH. What was the pH of the original benzoic acid solution?

### Pharmacy

Which of the following is a TRUE statement? Options: A. Because of the pharmacological processes of ethanol in the body, blood alcohol levels can decrease faster than they can rise. B. Because of the pharmacological processes of ethanol in the body, blood alcohol levels can rise faster than they can decrease. C. None of the answers are correct. D. Blood alcohol levels are not related to the pharmacological processes involving ethanol. E. Ethanol never leaves the body, but is used up as energy.

### Zoology

Primates can be grouped into two main categories. Which of the following correctly pairs the group with the location? Options: A. Old World: Asia, Africa and New World: South America B. Old World: Africa and New World: South America only C. New World: South America only D. Old World: Africa only E. Old World: Asia only

Figure 14: More cases of Open-Sci.

