# OpenReview forum: "Better know nothing than half-know anything: A Precise and Efficient Dataset for Scientific Reasoning in Language Models"
_ICLR.cc/2026/Conference — Submitted to ICLR 2026_

### Official Review · Reviewer_4dT2 · 2025-10-29

**Soundness:** 3
**Presentation:** 2
**Contribution:** 2
**Rating:** 4
**Confidence:** 4

**Summary:**

This paper presents PreciSci, a pipeline for constructing scientific reasoning datasets, and Open-Sci, a 196k dataset covering physics, chemistry, biology, and medicine. The pipeline extracts QA pairs from authoritative sources, formalizes questions, applies multi-stage filtering, and enforces answer consistency. Despite being 16% the size of MegaScience, Open-Sci reportedly achieves 4.49% better performance on scientific benchmarks.

**Strengths:**

1. The five-stage pipeline provides a structured methodology for scientific data curation with clear motivations for each component: authoritative source extraction, question formalization, noise mitigation, and answer consistency enforcement.

2. Comprehensive evaluation across 12 benchmarks spanning scientific domains (MedQA, PHYSICS, ChemBench, ProteinLMBench), general reasoning (GPQA, MMLU-Pro), and mathematics (AIME2025, Math-500), with consistent improvements across Qwen and Llama models at different scales.

3. Well-balanced dataset with 47 sub-disciplines, difficulty labels, diverse question types (48% open-ended, 30% calculation), and detailed answers averaging 1,124 tokens.

**Weaknesses:**

1. Bias from model-based filtering. Section 3.3 uses model accuracy thresholds for cascaded filtering, and Section 3.4 states "only correct candidates consistent with references were retained." This biases the dataset toward problems current models can solve, contradicting the goal of creating challenging training data for improving model capabilities.

2. Insufficient decontamination verification. Section 3.3 mentions using "embedding similarity and LLM confirming semantic overlap" for benchmark decontamination but provides no details: embedding model, similarity threshold, LLM prompts, or how many samples were removed per benchmark. Performance gains may result from data leakage rather than quality improvements.

3. Missing critical implementation details. No prompts for question formalization (Section 3.2), no numerical thresholds for filtering (Section 3.3), no method for answer extraction (Section 3.4), and no inter-annotator agreement statistics. The work is non-reproducible.

4. Limited methodological novelty. The pipeline combines standard techniques (LLM extraction, MinHash deduplication, answer distillation) without introducing new algorithms or theoretical insights.

**Questions:**

1. What does "only correct candidates consistent with references were retained" (Section 3.4) specifically mean? Are you filtering based on whether Qwen3-30B generates correct answers? What metric determines correctness, and what are the complete filtering criteria?

2. Provide complete decontamination specifications: which embedding model, what similarity threshold, what LLM prompts? For each of the 12 benchmarks, how many training samples were identified as overlapping and removed? Provide concrete examples.

3. How were the 200k samples selected from each dataset in Table 6? Random sampling, stratified sampling, or another method? What ensures fair comparison given different quality distributions?

4. The paper claims 196k samples are more efficient than 1.25M, but shows no scaling analysis. Can you provide results using different data amounts (e.g., 50k, 100k, 150k, 196k) to validate this claim?

5. Table 8 shows minimal difference between base Qwen3-8B (59.97%) and Open-Sci-tuned version (60.49%) in science reasoning. Given this negligible improvement, how do you justify the claim that Open-Sci significantly improves reasoning efficiency?

---

> ### Author Response · Authors · 2025-11-24
> **Response to Reviewer 4dT2 (1/8)**
>
> >W1: Bias from model-based filtering. Section 3.3 uses model accuracy thresholds for cascaded filtering, and Section 3.4 states "only correct candidates consistent with references were retained." This biases the dataset toward problems current models can solve, contradicting the goal of creating challenging training data for improving model capabilities.
>
> We agree that model-based filtering may discard some genuinely difficult items, but our inspection of rejected samples shows that such “hard” problems are heavily mixed with ambiguous, ill-posed, or internally inconsistent questions that can harm scientific-reasoning training. We need to create an inherent trade-off between retaining more potentially hard problems and aggressively filtering out harmful ones.
>
> To study this trade-off empirically under a fixed training budget, we relaxed the cascaded filtering and restored 25k previously removed items, mixing them with 25k clean Open-Sci samples, and compared this “mixed” 50k setting with training on 50k fully clean Open-Sci data. As shown in Table, the mixed dataset yields consistently lower performance across all benchmarks than the clean 50k setting.
>
> Table: evaluation results.
>
> | Benchmark        | Mixed (25k filtered + 25k clean) | Open-Sci (50k clean) |
> |------------------|----------------------------------|------------------------|
> | ChemBench        | 51.38                            | **53.64**              |
> | ProteinLMBench   | 56.14                            | **58.79**                 |
> | MedQA            | 75.26                            | **75.74**                 |
> | PHYSICS          | 31.97                            | **33.23**                 |
> | MMLU-Pro         | 68.28                            | **69.06**                 |
>
> This result indicates that, in our current regime, the negative impact of reintroducing noisy items outweighs the potential benefit of keeping additional hard cases.

---

> ### Author Response · Authors · 2025-11-24
> **Response to Reviewer 4dT2 (2/8)**
>
> >W2 && Q2: Provide complete decontamination specifications
>
>
> Thank you for raising the concern about decontamination and potential data leakage. We now provide full details of the process. For benchmark decontamination, all benchmark items and Open-Sci samples are encoded using bge-large-en-v1.5. For each training sample, we retrieve the top-5 nearest benchmark items and send them to an LLM judge (Llama-3.3-70B-Instruct, temperature = 0) for semantic-overlap verification. The prompt for LLM judge is updated in the appendix D. The number of removed items per benchmark is in the table.
>
> Table: number of removed items per benchmark.
>
> | Benchmark      | ProteinLMBench | PHYSICS | MedQA | PubMedQA | ChemBench | MMLU-Pro | SuperGPQA | GPQA | math | AIME-2025 |
> |----------------|----------------|---------|-------|----------|-----------|----------|-----------|------|------|-----------|
> | Nums           | 3115           | 4708    | 16821 | 191      | 6683      | 32313    | 52594     | 1741 | 354  | 0         |

---

> ### Author Response · Authors · 2025-11-24
> **Response to Reviewer 4dT2 (3/8)**
>
> >W3: Missing critical implementation details. No prompts for question formalization (Section 3.2), no numerical thresholds for filtering (Section 3.3), no method for answer extraction (Section 3.4), and no inter-annotator agreement statistics. The work is non-reproducible.
>
> Thank you for highlighting the missing implementation details and reproducibility concerns. We have now consolidated all key specifications into the revised appendix:
>
> - Question formalization: Below is a simplified version of prompt, we have updated the complete prompt in appendix D.
>
> ```### GLOBAL TASKS
> 1. **Strip embedded answers**:xxxxx
> 2. **Fix formatting & LaTeX**:xxxxx
> 3. xxxxxx
> ### PER-TYPE RULES
> Current Question Type: {question_type}
> #### 1. Multiple-Choice : xxxxx
> #### 2. True/False  : xxxxx
> #### 3. xxxx
> ```
>
> - Deduplication: we specify the numerical parameters for deduplication (MinHash + LSH with ngram = 3, similarity threshold = 0.6, b = 1, r = 10).
> - Cascaded filtering: We generate 8 rollouts for earch model and use an LLM-as-judge to decide whether each rollout is consistent with the reference answer. We use Qwen2.5-7B-Instruct as the smaller model, if the model answers correctly in more than 7 out of 8 rollouts (>7/8), the item is discarded as too easy. We use Qwen3-30B-A3B-Instruct as the stronger model. If all 8 rollouts are judged incorrect (0/8 correct), the item is discarded as likely low-quality or misaligned with the reference. The prompt of judgement has updated in teh appendix D.
> - Answer extraction: For each discipline, we design a subject-specific prompt to extract a reference answer from the source material. In the cascaded filtering stage, model predictions are then directly compared against this reference answer (after normalization).
> - Inter-annotator agreement statistics: We recruited 50 annotators (30 undergraduates and 20 graduate students). Annotators were allowed to use external tools (such as search engines) and followed a “quality-first” principle. Each item underwent triple-blind cross-validation, and only samples with over 50% agreement were retained.
>
> In addition, a subset of Open-Sci is available in supplementary material. The full data, models and code will be released. With these additions, we hope the pipeline is fully reproducible.

---

> ### Author Response · Authors · 2025-11-24
> **Response to Reviewer 4dT2 (4/8)**
>
> >W4: Limited methodological novelty. The pipeline combines standard techniques (LLM extraction, MinHash deduplication, answer distillation) without introducing new algorithms or theoretical insights.
>
> Thank you for raising this point. While our work does not propose a new training objective or backbone architecture, it is not a simple reuse of standard filtering tools. The methodological contribution lies in how we reformulate and integrate these components into a precision-first pipeline specifically for scientific reasoning, along two axes: (a). Precision-first, difficulty-aware curation for scientific reasoning. (b). Multi-stage verification pipeline combining lexical, embedding, and LLM-as-judge signals. We construct a small but highly effective dataset that substantially improves scientific-reasoning performance. This precision-first curation approach is the main conceptual contribution of our work.

---

> ### Author Response · Authors · 2025-11-24
> **Response to Reviewer 4dT2 (5/8)**
>
> >Q1: （1）What does "only correct candidates consistent with references were retained" (Section 3.4) specifically mean?（2） Are you filtering based on whether Qwen3-30B generates correct answers? （3）What metric determines correctness, and what are the complete filtering criteria?
>
> (1). This phrase refers to the stage of our cascaded filtering, where each candidate QA pair must pass an explicit correctness check against a reference answer. For each question, we require that the model can produce at least one answer that is judged semantically consistent with this reference; only such “correct” candidates are retained, and all others are discarded.
>
> (2). Yes. In this stage, filtering is indeed based on whether Qwen3-30B-A3B-Instruct can generate a correct answer relative to the reference. We sample multiple rollouts from Qwen3-30B-A3B-Instruct for each question.
>
> (3). Correctness is determined by an LLM-as-judge procedure, again using Qwen3-30B-A3B-Instruct with a separate judging prompt. For each rollout, the judge compares the model’s answer with the reference answer and decides whether they are semantically consistent (correct vs. incorrect). We have updated the prompt for judgement in the appendix D.

---

> ### Author Response · Authors · 2025-11-24
> **Response to Reviewer 4dT2 (6/8)**
>
> >Q3: How were the 200k samples selected from each dataset in Table 6? Random sampling, stratified sampling, or another method? What ensures fair comparison given different quality distributions?
>
> We use simple random sampling when selecting the 200k samples for each dataset. Random sampling preserves the original distribution and avoids injecting any additional bias that stratification or heuristic filtering might introduce. To verify that the comparison is fair and that random sampling is stable, we randomly drew three independent 200k subsets from MegaScience and trained models under identical settings. As shown in Table, the performance across the three runs is highly consistent, demonstrating that random sampling yields stable results and does not depend on a particular subset realization.
>
> Table: the results of stability verification.
>
> | Benchmark        | Sample-1 (200k) | Sample-2 (200k) | Sample-3 (200k) | Mean   | Std  |
> |------------------|------------------|------------------|------------------|--------|------|
> | ChemBench        | 53.18           | 53.12           | 52.81           | 53.04 | 0.20 |
> | ProteinLMBench   | 55.61           | 54.24           | 54.98           | 54.94 | 0.69 |
> | MedQA            | 73.14           | 72.85           | 73.46           | 73.15 | 0.31 |
> | PHYSICS          | 26.44           | 26.21           | 27.13           | 26.59 | 0.48 |
> | MMLU-Pro         | 64.41           | 63.88           | 64.11           | 64.13 | 0.27 |

---

> ### Author Response · Authors · 2025-11-24
> **Response to Reviewer 4dT2 (7/8)**
>
> >Q4: The paper claims 196k samples are more efficient than 1.25M, but shows no scaling analysis. Can you provide results using different data amounts (e.g., 50k, 100k, 150k, 196k) to validate this claim?
>
> Thank you for the suggestion. We conducted a scaling analysis using subsets of 0k, 10k, 50k, 100k, 150k, and 195k samples. As shown in the table below, the average performance increases consistently with data size, confirming the efficiency of Open-Sci. The gain becomes smaller after ~150k, but the trend remains positive up to 195k.
>
> This supports our claim that Open-Sci achieves strong performance with substantially fewer samples, while still benefiting from further scaling.
>
> Table: scaling trends.
>
> |Training data size | Avg Performance | Trend |
> |----------:|----------------:|:-----|
> | 0         | 51.64 | ▇▇▇ |
> | 10k       | 54.96 | ▇▇▇▇▇▇ |
> | 50k       | 55.37 | ▇▇▇▇▇▇▇ |
> | 100k      | 56.05 | ▇▇▇▇▇▇▇▇ |
> | 150k      | 56.11 | ▇▇▇▇▇▇▇▇▇ |
> | 195k      | 56.29 | ▇▇▇▇▇▇▇▇▇▇ |

---

> ### Author Response · Authors · 2025-11-24
> **Response to Reviewer 4dT2 (8/8)**
>
> >Q5: Table 8 shows minimal difference between base Qwen3-8B (59.97%) and Open-Sci-tuned version (60.49%) in science reasoning. Given this negligible improvement, how do you justify the claim that Open-Sci significantly improves reasoning efficiency?
>
> The main goal of our work is to improve scientific reasoning under instruction-following settings with high-precision data. Table 8 was originally included to show that even when evaluated in “thinking” mode, precision-cleaned instruction data can slightly improve performance while reducing token usage.
>
> We agree that the improvement in the earlier version of Table 8 was not sufficiently clear. Following the reviewer’s suggestion, we extended the training for an additional 2 epochs under the same settings. The updated results show a larger performance improvement and a substantial reduction in reasoning tokens. This confirms our claim that more precise data leads to more efficient reasoning. The model requires fewer tokens to reach higher accuracy. We have updated Table 8 accordingly in the revised manuscript.
>
> Table: comparison of reasoning efficiency.
>
> | Model                 | Tokens | Science Avg. | General Avg. |
> |-----------------------|--------|--------------|--------------|
> | Qwen3-8B-Think        | 3262   | 59.97        | 58.34        |
> | Qwen3-8B-OpenSci-Think | 2164   | 61.03        | 59.74        |

---

### Official Review · Reviewer_Rf7n · 2025-10-31

**Soundness:** 3
**Presentation:** 3
**Contribution:** 3
**Rating:** 6
**Confidence:** 3

**Summary:**

This paper introduces "PreciSci," a meticulous data curation pipeline designed to address the poor quality and noise prevalent in existing scientific reasoning datasets. The authors argue for a "quality over quantity" approach, starting from authoritative sources (textbooks, exams) and applying rigorous multi-stage filtering that ultimately discards over 67% of the initial raw data.
The resulting dataset, "Open-Sci," contains 196k high-precision, knowledge-dense instances. Experiments show that models fine-tuned on Open-Sci, despite its small size (1/6th of MegaScience), achieve superior performance, improving by an average of 4.49% on scientific benchmarks  and demonstrating remarkable generalization to mathematics.

**Strengths:**

1. Clear Motivation: The paper correctly identifies data quality, not just scale, as the primary bottleneck for scientific reasoning. The PreciSci pipeline is a robust and well-documented methodology for tackling this.
2. Exceptional Generalization (Math): The most striking result is the massive performance leap on mathematical reasoning. For example, Qwen3-8B's score on AIME2025 jumps from a baseline of 16.67% to 51.67% when trained on Open-Sci. This strongly suggests that rigorous, multi-step scientific training imparts transferable reasoning skills.
3. Thorough Ablation Studies: The authors demonstrate that each component of the PreciSci pipeline (formalization, noise mitigation, answer consistency) is critical to the final performance.

**Weaknesses:**

1. Analysis of Math Gains is Lacking: The paper does not adequately investigate why the AIME2025 performance gain is so exceptionally large (+30 points). A deeper analysis correlating the 30% "calculation" questions in Open-Sci to the structure of math problems is needed.
2. Unquantified Human Cost: The pipeline relies on a "hybrid AI-human process", but the human-in-the-loop effort is not quantified. This makes it difficult to assess the true cost and reproducibility of the PreciSci pipeline.
3. Potential Pipeline Bias: The pipeline itself uses Qwen and DeepSeek models for key steps like QA extraction, formalization, and answer generation. This raises a concern: Does the Open-Sci dataset inadvertently "overfit" to the style of Qwen models? This could explain why Qwen models show significantly larger gains (+8.69% for 8B) from Open-Sci than other models like Llama-3.1 (+3.11%).

**Questions:**

see weakness

---

> ### Author Response · Authors · 2025-11-24
> **Response to Reviewer Rf7n (1/3)**
>
> >W1: Analysis of Math Gains is Lacking: The paper does not adequately investigate why the AIME2025 performance gain is so exceptionally large (+30 points). A deeper analysis correlating the 30% "calculation" questions in Open-Sci to the structure of math problems is needed.
>
> The "calculation" subset implicitly force the model to use mathematics as a backbone for scientific reasoning. We provide a focused quantitative analysis of the “calculation” subset. A random sample of 2k problems annotated with GPT-5 shows that 98.8% require non-trivial mathematical reasoning, and ~9–10% require mathematics as a critical step for completing the reasoning process.
>
> We further categorize the required mathematical skills and find broad coverage across algebraic equation solving (81.8%), dimensional analysis (76.2%), calculus-style rate/accumulation reasoning (20.8%), scaling/logarithmic behaviors (12.6%), and others.
>
> Although Open-Sci is not a math dataset, these calculation problems systematically train quantitative reasoning skills that directly transfer to math. We believe this provides a natural and plausible explanation for the strong AIME2025 gains.

---

> ### Author Response · Authors · 2025-11-24
> **Response to Reviewer Rf7n (2/3)**
>
> >W2: Unquantified Human Cost: The pipeline relies on a "hybrid AI-human process", but the human-in-the-loop effort is not quantified. This makes it difficult to assess the true cost and reproducibility of the PreciSci pipeline.
>
> During the construction of Open-Sci, we first apply model-based filters to extract and clean candidate QA pairs. Only a subset of items—those flagged as low-confidence—is routed to human annotators for verification and keep/discard decisions. To validate these candidate samples, we recruited **50** annotators (30 undergraduates and 20 graduate students) over a **two-week** period. Annotators were allowed to use external tools (such as search engines) and followed a “quality-first” principle. This hybrid design follows the practices of prior large-scale datasets [1, 2], which rely primarily on automatic filters supplemented by targeted manual checks rather than fully manual annotation. Under current resource constraints, we believe that this process represents a reasonable compromise between data quality and human cost, and that we have made every reasonable effort; future users can further refine the data based on their own resources and funds.
>
> [1] MDBench: A Synthetic Multi-Document Reasoning Benchmark Generated with Knowledge Guidance
>
> [2] mCSQA: Multilingual Commonsense Reasoning Dataset with Unified Creation Strategy by Language Models and Humans

---

> ### Author Response · Authors · 2025-11-24
> **Response to Reviewer Rf7n (3/3)**
>
> >W3: Potential Pipeline Bias: The pipeline itself uses Qwen and DeepSeek models for key steps like QA extraction, formalization, and answer generation. This raises a concern: Does the Open-Sci dataset inadvertently "overfit" to the style of Qwen models? This could explain why Qwen models show significantly larger gains (+8.69% for 8B) from Open-Sci than other models like Llama-3.1 (+3.11%).
>
> We appreciate the reviewer’s concern. Our analyses indicate that the gains from Open-Sci do not primarily arise from stylistic alignment with Qwen models, but from the underlying data quality. To directly test whether the pipeline biases the dataset toward Qwen-style outputs, we conducted two controlled experiments:
>
> 1. Same distillation test.
>
>     We took 200k samples from MegaScience and used Qwen3-30B-A3B-Instruct to distill. Under identical model distillation and data size settings, Open-Sci still yields significantly higher performance across all benchmarks. This suggests that the improvements are not explained by stylistic or architectural alignment with Qwen, but by the higher precision and better curation of Open-Sci.
>
>     Table: evaluation results for same distillation test.
>
>     | Benchmark        | MegaSci. | Ours   |
>     |------------------|----------|--------|
>     | ChemBench        | 48.39    | 55.28  |
>     | ProteinLMBench   | 56.14    | 58.69  |
>     | MedQA            | 76.53    | 77.43  |
>     | PHYSICS          | 33.88    | 36.75  |
>     | MMLU-Pro         | 69.15    | 70.48  |
>
> 2. Cross-family evaluation with Llama.
>
>     We further distilled the same data from Llama-3.1-70B-Instruct and fine-tuned Llama-3.1-8B. In this setting, the Llama-distilled variant sometimes achieves slightly higher scores on ChemBench, but training directly on Open-Sci still yields the strongest results on most benchmarks. This pattern suggests that different teacher models emphasize different aspects of scientific reasoning, and that a hybrid or mixed-source distillation strategy may further improve Llama-family students. We appreciate the reviewer’s comment, which directly led us to this additional analysis and revealed a potentially useful future direction.
>
>     Table: evaluation results for Llama.
>
>     | Benchmark        | Llama-3.1-8B | Llama-8B-Distill-70B | Ours   |
>     |------------------|--------------|------------------------|--------|
>     | ChemBench        | 40.14        | **50.41**                  | 40.94  |
>     | ProteinLMBench   | 55.40        | 54.66                  | **57.52**  |
>     | MedQA            | 65.60        | 66.61                  | **67.99**  |
>     | PHYSICS          | 7.65         | 11.67                  | **15.13**  |
>     | MMLU-Pro         | 48.08        | 52.56                  | **54.91**  |

---

### Official Review · Reviewer_Qsbw · 2025-10-31

**Soundness:** 3
**Presentation:** 3
**Contribution:** 2
**Rating:** 4
**Confidence:** 5

**Summary:**

This paper introduces PreciSci, a data curation pipeline designed to construct high-quality, compact scientific reasoning datasets, and Open-Sci, a 196k-instance dataset built using this pipeline. The core idea is that precision and quality outweigh scale in scientific reasoning.

The pipeline involves:
1. Extracting QA pairs from authoritative sources (textbooks, exams, competitions) using a hybrid AI-human approach;
2. Formalizing questions into standard types (MCQ, fill-in, calculation, etc.) for clarity and completeness;
3. Multi-stage noise mitigation (deduplication, decontamination, filtering) to remove redundancy and low-quality samples;
4. Answer consistency refinement via distillation to ensure accurate, well-structured reasoning.

**Strengths:**

S1: This paper addresses a real gap—lack of high-quality scientific reasoning data—and proposes a quality-over-quantity approach, which is refreshing in the era of scaling.

S2: PreciSci is well-structured, with clear stages for extraction, formalization, filtering, and answer refinement. The ablation studies show each step contributes meaningfully.

S3: Open-Sci consistently outperforms larger datasets like MegaScience and WebInstruct across scientific, general, and math benchmarks, even on smaller models.

S4: This paper demonstrates that 196k well-curated samples can beat 1.2M+ noisy ones, supporting the paper’s central thesis.

**Weaknesses:**

W1: In general, STEM includes the math domain. However, Open-Sci is limited to natural sciences and excludes math, which is a major part of scientific reasoning. The claim of “scientific reasoning” is thus partially incomplete.

W2: Despite claiming human-in-the-loop, there is no quantitative human evaluation of question quality, answer correctness, or reasoning depth.

W3: The paper does not test generalization to new domains or out-of-distribution scientific problems. Take Table 4 as an example, SuperGPQA and GPQA-Diamond are also benchmarks for scientific reasoning, not general domains.

W4: This paper only compares to other datasets, not to models trained with process supervision, self-generated data, or RL-based reasoning, which are more relevant in 2025.

W5: This paper would be valuable for the community if the dataset, models, and pipeline were fully open-sourced and more examples were provided to learn about the quality of Open-Sci.

**Questions:**

Q1: How does Open-Sci perform on unseen/evaluated scientific domains (e.g., astronomy, geology) or interdisciplinary problems? For example, providing the result of each subject of MMLU-Pro should be better.

Q2: What is the actual human effort involved? How many hours or annotators were used, and what was the inter-annotator agreement?

Q3: How does the model perform on problems requiring creative or open-ended reasoning, not just standard QA formats?

Q4: Would Open-Sci still outperform larger datasets if both were scaled to the same size? Is the gain purely due to quality, or also due to better coverage?

Q5: How does the pipeline handle ambiguous or ill-posed questions that are common in real-world scientific inquiry?

Q6: Can the pipeline be automated further to reduce human involvement, or is it inherently labor-intensive?

---

> ### Author Response · Authors · 2025-11-24
> **Response to Reviewer Qsbw (1/11)**
>
> >W1: In general, STEM includes the math domain. However, Open-Sci is limited to natural sciences and excludes math, which is a major part of scientific reasoning. The claim of “scientific reasoning” is thus partially incomplete.
>
> Thank you for raising this important point. Our work intentionally focuses on natural sciences, physics, chemistry, biology, and medicine, rather than the broader STEM spectrum. This scope choice follows the dataset’s stated goal of providing precise, knowledge-grounded scientific reasoning data derived from authoritative sources in the natural sciences.
>
> We agree that mathematical reasoning is closely intertwined with scientific reasoning, especially in physics and quantitative domains. To evaluate whether incorporating mathematics into our Open-Sci dataset improves performance, we conducted an additional controlled experiment by adding 50k math samples from MegaScience and NaturalReasoning (balanced to preserve domain proportions).
>
> Fine-tuning Qwen3-8B on this math-augmented dataset yields small but notable gains on physics (**+2.45%**), MMLU-Pro (**+0.67%**), but slight regressions on biology (**−0.95%**) and some other natural-science benchmarks. These mixed results suggest that mathematical data can indeed strengthen certain forms of scientific reasoning, but naively mixing in heterogeneous math sources weaken performance in areas (such as biology) that are less directly tied to those math distributions.
>
> We therefore see rigorous mathematical extensions of Open-Sci—using a version of our PreciSci pipeline tailored to mathematical content and discipline balance—as an important direction for future work. Our current design intentionally leaves room for such an extension, while this paper focuses on demonstrating the benefits of a high-precision, natural-science–centered core.
>
> Table: Impact of integrating 50k math samples into Open-Sci.
>
> | Model              | ChemBench | ProteinLMBench | MedQA | PHYSICS | MMLU-Pro |
> |--------------------|-----------|----------------|-------|---------|----------|
> | Qwen3-8B-Instruct  | 52.36     | 57.63          | 75.33 | 25.17   | 65.52    |
> | Ours      | **54.33** | **58.26**      | **76.84** | 31.66 | 68.91 |
> | Ours+add math | 54.21 | 57.31         | 76.74 | **34.11** | **69.58** |

---

> ### Author Response · Authors · 2025-11-24
> **Response to Reviewer Qsbw (2/11)**
>
> >W2: Despite claiming human-in-the-loop, there is no quantitative human evaluation of question quality, answer correctness, or reasoning depth.
>
> To identify and filter out low-quality scientific QA pairs, we adopt a rubric-based evaluation protocol following previous work [1]. Specifically, for each QA pair, annotators assign 1–5 scores on five dimensions using a fixed rubric: (1) **question clarity** (how well-posed and unambiguous the question is), (2) **scientific relevance** (whether it clearly belongs to natural science), (3) **reasoning difficulty** (from simple factual recall to multi-step reasoning), (4) **QA semantic consistency** (whether the answer directly and fully addresses the question), and (5) **factual correctness** (the scientific accuracy of the answer).
>
> To align with the reviewer’s three dimensions, we derive aggregate criteria from these scores:
> - High-quality questions if clarity_score + scientific_relevance_score > 5
> - Correct answers if qa_consistency_score + factual_correctness_score > 5
> - Non-trivial reasoning if difficulty_score >= 2.
> Using this rubric, we performed a human study on 200 randomly sampled QA pairs, annotated by 10 annotators (8 undergraduates, 2 graduates). The results are:
>
> | High-quality questions              | Correct answers | Non-trivial reasoning required  |
> |-----------|-----------|-----------|
> | 98%  | 94%     | 98%          |
>
> Since our thresholds jointly require clarity and domain relevance for questions, semantic alignment and factual correctness for answers, and at least moderate reasoning for non-triviality, these numbers indicate that the vast majority of Open-Sci items are clear, scientifically grounded, and correctly answered, and that almost all of them demand more than simple recall.
>
> [1]QGEval: A Benchmark for Question Generation Evaluation

---

> ### Author Response · Authors · 2025-11-24
> **Response to Reviewer Qsbw (3/11)**
>
> >W3 && Q1 ...providing the result of each subject of MMLU-Pro should be better...
>
> Following your recommendation, we now report the full per-subject results on MMLU-Pro, which includes many unseen or lightly represented scientific and interdisciplinary fields such as engineering(**+10.01%**), economics(**+2.61%**), psychology(**+1.38%**), philosophy(**+5.81%**), and computer science(**+5.61%**). Open-Sci improves across nearly all OOD subjects which shows that high-precision scientific reasoning data enhances generalizable reasoning skills, not merely domain-specific recall.
>
> Following your suggestion, we have updated the appendix C with the complete per-discipline results, including detailed scores for MMLU-Pro sub-subjects evaluations.
>
> Table: unseen discipline results on MMLU-Pro.
>
> | Model             | Math   | Law    | Engineering | Economics | Psychology | Business | Philosophy | CS    | History | Other |
> |-------------------|--------|--------|-------------|-----------|------------|----------|------------|-------|---------|-------|
> | Qwen3-8B          | 83.94  | 34.51  | 52.32       | 74.64     | 69.67      | 73.64    | 54.51      | 69.51 | 53.54   | 58.87 |
> | Qwen3-8B-MegaSci. | 83.86  | 37.42  | 55.21       | 75.95     | **71.55**  | 73.13    | 58.32      | 67.32 | 55.91   | **63.64** |
> | Ours              | **90.67** | **38.24** | **62.33**   | **77.25** | 71.05      | **77.69** | **60.32**  | **75.12** | **56.96** | 63.31 |

---

> ### Author Response · Authors · 2025-11-24
> **Response to Reviewer Qsbw (4/11)**
>
> >W4: This paper only compares to other datasets, not to models trained with process supervision, self-generated data, or RL-based reasoning, which are more relevant in 2025.
>
> Thank you for raising this important point. We fully agree that process supervision, self-generated data, and RL-based reasoning are central methodological directions in 2025.
>
> (a). Our contribution in this paper is on data curation: we construct and analyze a high-precision scientific-reasoning dataset, and our main comparisons therefore focus on different data sources and curation strategies under standard supervised finetuning, rather than on proposing or benchmarking new RL or process-supervision algorithms.
>
> (b). To verify that Open-Sci can also support modern RL-style reasoning training, we conducted a pilot RL experiment on Qwen3-8B. We selected a balanced 4k subset of Open-Sci and applied the DAPO algorithm with outcome, format, and soft overlong rewards. Training used a learning rate of 1e-6, a global batch size of 64, 16 sampled generations per prompt, and CompassVerifier-7B for reward scoring.
>
> As summarized in the table, the RL-trained model consistently outperforms the base model on both scientific and general benchmarks (e.g., **+2.22%** on ChemBench, **+3.47%** on PHYSICS).
>
> These results demonstrate that (i) Open-Sci’s high-precision supervision is fully compatible with modern RL-style reasoning training, and (ii) even a small-scale, direct RL finetuning on Open-Sci can further improve scientific and general reasoning performance beyond the base model.
>
> This suggests that RL-based reasoning on Open-Sci is itself a promising direction for systematic future study. In this paper, we therefore focus on providing a precise, scalable scientific dataset, while larger-scale RL, richer process-level rewards, and broader comparisons across RL algorithms on Open-Sci are left as important future work.
>
> Table: evaluation of RL-based model.
>
> | Model              | ChemBench | ProteinLMBench | MedQA | PHYSICS | MMLU-Pro |
> |--------------------|-----------|----------------|-------|---------|----------|
> | Qwen3-8B-Instruct  | 52.36     | 57.63          | 75.33 | 25.17   | 65.52    |
> | Ours-RL      | **54.58** | **57.94**      | **76.57** | **28.64** | **65.72** |

---

> ### Author Response · Authors · 2025-11-24
> **Response to Reviewer Qsbw (5/11)**
>
> >W5: This paper would be valuable for the community if the dataset, models, and pipeline were fully open-sourced and more examples were provided to learn about the quality of Open-Sci.
>
> We agree that openness is crucial for enabling the community to benefit from scientific–reasoning datasets. We have already released a subset of Open-Sci in the supplementary materials. And we will release the full dataset, the trained models and the code, ensuring that the main results of our work are fully reproducible. To further illustrate the quality of Open-Sci, we have added more representative examples across different disciplines. These can be found in Appendix E.
> We hope that these additions address the reviewer’s concern and significantly enhance the potential value of our work to the community.

---

> ### Author Response · Authors · 2025-11-24
> **Response to Reviewer Qsbw (6/11)**
>
> >Q2: What is the actual human effort involved? How many hours or annotators were used, and what was the inter-annotator agreement?
>
> We recruited **50** annotators (30 undergraduates and 20 graduate students) over a **two-week** period. Annotators were allowed to use external tools (such as search engines) and followed a “quality-first” principle. Each item underwent **triple-blind cross-validation**, and only samples with **over 50% agreement** were retained.

---

> ### Author Response · Authors · 2025-11-24
> **Response to Reviewer Qsbw (7/11)**
>
> >Q3: How does the model perform on problems requiring creative or open-ended reasoning, not just standard QA formats?
>
> Open-Sci is not designed to enhance creativity-oriented or open-ended reasoning; instead, it explicitly targets precise, well-specified scientific problems with scientifically verifiable answers. This differs substantially from creative or open-ended reasoning tasks, which emphasize divergent thinking rather than scientific rigor. Because of this domain mismatch, we do not expect strong transfer from Open-Sci to creativity-centered benchmarks, and we did not position our dataset as a solution for such tasks.
>
> We agree that open-ended or creative scientific reasoning is an important and interesting direction. Building on our precision-oriented pipeline, we see it as natural future work to extend Open-Sci toward tasks such as hypothesis generation, multi-perspective scientific explanation, which would broaden the scope from rigorous problem solving toward more general forms of scientific intelligence.

---

> ### Author Response · Authors · 2025-11-24
> **Response to Reviewer Qsbw (8/11)**
>
> `Q4: （1）Would Open-Sci still outperform larger datasets if both were scaled to the same size?`
>
> As shown in the table below, when scaled using data of the same high quality, Open-Sci still outperforms larger datasets. Specifically, we study the scaling trend of Open-Sci by training with subsets of increasing size (0 → 195k). The average performance improves monotonically as the number of Open-Sci samples increases (from 51.64% to 56.29%). This suggests that if Open-Sci were scaled to the same size as other corpora, its performance would likely increase further.
>
> Table: scaling trend.
>
> | Training data size | Avg Performance | Trend |
> |----------:|----------------:|:-----|
> | 0         | 51.64 | ▇▇▇ |
> | 10k       | 54.96 | ▇▇▇▇▇▇ |
> | 50k       | 55.37 | ▇▇▇▇▇▇▇ |
> | 100k      | 56.05 | ▇▇▇▇▇▇▇▇ |
> | 150k      | 56.11 | ▇▇▇▇▇▇▇▇▇ |
> | 195k      | 56.29 | ▇▇▇▇▇▇▇▇▇▇ |

---

> ### Author Response · Authors · 2025-11-24
> **Response to Reviewer Qsbw (9/11)**
>
> `Q4: (2) Is the gain purely due to quality, or also due to better coverage?`
>
> We suggest that the improvement is primarily driven by data quality. Once quality is ensured in the constructed dataset, a broader discipline coverage can further benefit the performance.
>
> - Quality matters：
>
> (a). To directly test whether scaling benefits depend on data quality, we relaxed our cascaded filtering and restored 25k low-quality samples, forming a (i) Noisy-Subset (50k) setting(25k low-quality + 25k random Open-Sci) and compare it with a (ii) High-Quality (50k) setting(50k random Open-Sci). Results show that High-Quality(50k) clearly outperforms both the baseline and the Noisy setting. Furthermore, Noisy-Subset, despite having the same size, performs noticeably worse—even dropping below the baseline on several benchmarks. These phenomena verify the gain driven by data quality.
>
> (b). To isolate data quality from disciplinary coverage, we performed a single-discipline experiment: (i) Qwen3-8B-MegaSci-Med-50k: 50k Medicine samples from MegaScience. (ii). Open-Sci-Med-50k: 50k Medicine samples from Open-Sci. Open-Sci-Med outperforms MegaSci-Med across most scientific benchmarks. With identical size and identical domain coverage, this confirms that Open-Sci’s advantage comes from data quality.
>
> Table: evaluation results.
>
> | Benchmark        | Qwen3-8B | Noisy-Subset | High-Quality | MegaSci-Med-50k | Open-Sci-Med-50k |
> |------------------|----------|--------------|--------------|------------------|-------------------|
> | ChemBench        | 52.36    | 51.38        | 53.64        | 49.95            | 52.53             |
> | ProteinLMBench   | 57.63    | 56.14        | 58.79        | 60.70            | 58.37             |
> | MedQA            | 75.33    | 75.26        | 75.74        | 71.27            | 76.93         |
> | PHYSICS          | 25.17    | 31.97        | 33.23        | 20.66            | 29.91             |
> | MMLU-Pro         | 65.52    | 69.28        | 69.06        | 61.55            | 68.83             |
>
>
> - Broader discipline coverage further benefits.
>
> We next studied the effect of coverage on our Open-Sci: (i). Open-Sci-Med-50k: single-discipline. (ii). Open-Sci-Mixed-50k: 50k samples randomly from Open-Sci with full disciplines. It shows that expanding the coverage further improves the performance of the model.
>
>
> Table: evaluation results.
>
> | Benchmark        | Qwen3-8B | Open-Sci-Med-50k | Open-Sci-Mixed-50k |
> |------------------|----------|-------------------|----------------------|
> | ChemBench        | 52.36    | 52.53             | **53.64**               |
> | ProteinLMBench   | 57.63    | 58.37             | **58.79**               |
> | MedQA            | 75.33    | **76.93**         | 75.74               |
> | PHYSICS          | 25.17    | 29.91             | **33.23**               |
> | MMLU-Pro         | 65.52    | 68.83             | **69.06**              |

---

> ### Author Response · Authors · 2025-11-24
> **Response to Reviewer Qsbw (10/11)**
>
> >Q5: How does the pipeline handle ambiguous or ill-posed questions that are common in real-world scientific inquiry?
>
> To remove ambiguous or ill-posed questions, we define a taxonomy covering general language ambiguities (missing information, multiple interpretations, subjective terms, ill-formed notation) and scientific ambiguities (violations of scientific laws, conceptual misuse, unphysical parameters, ill-posed models). A strong LLM first flags these cases with confidence scores, and all low-confidence items are verified and filtered by humans. Then ambiguous or ill-posed questions were directly deleted.
>
> To assess the necessity of this step, we performed an ablation: training on 25k Open-Sci + 25k filtered ambiguous questions leads to significant performance drops compared to training on 50k clean Open-Sci. This confirms that ambiguous or ill-posed questions severely harm scientific reasoning quality, and that our filtering stage is essential.
>
> Table: evaluation results.
>
> | Benchmark        | Mixed (25k filtered + 25k clean) | Open-Sci (50k clean) |
> |------------------|----------------------------------|------------------------|
> | ChemBench        | 44.24                            | **53.64**                 |
> | ProteinLMBench   | 55.93                            | **58.79**                 |
> | MedQA            | 72.46                            | **75.74**                 |
> | PHYSICS          | 25.92                            | **33.23**                 |
> | MMLU-Pro         | 65.69                            | **69.06**                 |

---

> ### Author Response · Authors · 2025-11-24
> **Response to Reviewer Qsbw (11/11)**
>
> >Q6: Can the pipeline be automated further to reduce human involvement, or is it inherently labor-intensive?
>
> Our pipeline is already mostly automated. Human effort is used for quality control in low-confidence cases. Since these human decisions are structured, they can be used to train a specialized data-quality judge model, which would further reduce human involvement in future iterations. The pipeline is therefore not inherently labor-intensive, and its automation can be expanded naturally.

---

### Author Response · Authors · 2025-12-03
**General Response for new AC**

We thank the reviewers (R_Qsbw, R_Rf7n, R_4dT2) for their constructive feedback and for recognizing the value of our "quality-over-quantity" approach (R_Qsbw, R_Rf7n), the structural soundness of the PreciSci pipeline (R_Qsbw, R_4dT2), and the strong performance of Open-Sci (R_Qsbw, R_Rf7n, R_4dT2).

We have uploaded a revised manuscript incorporating the reviewers' suggestions, including expanded implementation details, prompt definitions, and deeper ablation studies. Below, we summarize our response to the shared concerns regarding decontamination, the source of reasoning gains, reproducibility, the data difficulty–quality trade-off and scaling behavior. Please find our detailed, **point-by-point responses** in the **individual replies** to each reviewer. All newly added or revised content in the manuscript is clearly highlighted in red for ease of review.

> Explaining the "Math Transfer" & Model Generalization (Addressing R_Qsbw, R_Rf7n)

Reviewers noted the great strength of Open-Sci on mathematical benchmarks (e.g., AIME 2025), despite the dataset being focused on natural sciences.

We analyzed the dataset and found that **30%** of Open-Sci consists of calculation-heavy scientific problems (e.g., stoichiometry, kinematics). A granular analysis (a random sample of 2k problems annotated with GPT-5) shows that **98.8%** of these calculation problems require non-trivial mathematical reasoning. We further categorize the required mathematical skills and find broad coverage across algebraic equation solving (**81.8%**), dimensional analysis (**76.2%**), calculus-style rate/accumulation reasoning (**20.8%**), scaling/logarithmic behaviors (**12.6%**). This explains the transfer: many physics and chemistry questions in Open-Sci instantiate the same core mathematical competencies evaluated by standalone math benchmarks.

> Reproducibility and Transparency (Addressing R_Qsbw, R_4dT2)

We are committed to full open science. In the revised manuscript and in our detailed responses to the reviewers, we have added (1) **Full Prompt Engineering**: The exact prompts for Question Formalization, Answer Extraction, and Decontamination (Appendix D & Figures 7-13). (2) **Human Annotation Details**: Specification of the 50-annotator team (30 undergrads, 20 grads), the triple-blind cross-validation process, and the resulting inter-annotator agreement standards. (3) **Resource Release**: We have released a subset of the data in the supplementary material on OpenReview and commit to releasing the full **Open-Sci dataset, the trained models, and the curation code** upon acceptance.

> The Trade-off: Difficulty vs. Noise (Addressing R_4dT2)

R3 questioned whether filtering "incorrect" model outputs removes valuable "hard" samples.

We conducted a specific ablation where we reintroduced 25k "hard/noisy" samples that were previously filtered out. The model trained on this "Mixed" dataset performed **consistently worse** (e.g., **-2.26%** on ChemBench, **-2.65%** on ProteinLMBench and **-1.26%** on PHYSICS) than the model trained on the clean Open-Sci subset. It shows that the "hard" problems are heavily mixed with ambiguous, ill-posed, or internally inconsistent questions that can harm scientific-reasoning training. Our filtering strategy effectively optimizes the signal-to-noise ratio, facilitating the performance of trained models.

> Scaling Behavior & Efficiency Claims (Addressing R_4dT2, R_Qsbw)

We conducted a scaling analysis using subsets of 0k, 10k, 50k, 100k, 150k, and 195k samples. As shown in the table below, the average performance increases consistently with data size, confirming the efficiency of Open-Sci. The gain becomes smaller after ~150k, but the trend remains positive up to 195k.

This supports our claim that Open-Sci achieves strong performance with substantially fewer samples, while still benefiting from further scaling.

Table: scaling trends.

| Training data size | Avg Performance | Trend |
|----------:|----------------:|:-----|
| 0         | 51.64 | ▇▇▇ |
| 10k       | 54.96 | ▇▇▇▇▇▇ |
| 50k       | 55.37 | ▇▇▇▇▇▇▇ |
| 100k      | 56.05 | ▇▇▇▇▇▇▇▇ |
| 150k      | 56.11 | ▇▇▇▇▇▇▇▇▇ |
| 195k      | 56.29 | ▇▇▇▇▇▇▇▇▇▇ |

---

### Meta-Review · Area_Chair_69KW · 2025-12-23

**Summary:**

This submission proposes PreciSci, a multi-stage curation pipeline, and Open-Sci, a 196k “quality over quantity” scientific reasoning dataset.

Reviewers agreed the motivation is timely and the empirical results are strong across multiple science benchmarks, with notable transfer to math.

The main concerns that informed the decision were:

(1) **insufficient implementation transparency** and **decontamination details** in the original draft, creating risk of leakage and limiting reproducibility,

(2) **potential bias** introduced by model-based filtering that may preferentially retain problems current models can already solve, and

(3) **limited methodological novelty** beyond combining established curation components. Additional concerns included incomplete quantification of human effort, and the practical value hinging on full release of data and code.

**Reviewer Concerns:**

**Concerns largely addressed by the rebuttal:**

* Reproducibility gaps were meaningfully improved via added prompts, filtering thresholds, sampling details, and clearer descriptions of the pipeline stages.
* Decontamination was clarified with the embedding model used, the retrieval-then-LLM-judge procedure, and counts of removed items per benchmark.
* The paper added scaling evidence over multiple dataset sizes, supporting the efficiency claim.
* The rebuttal provided a more direct analysis for the large math transfer and quantified human involvement and rubric-based spot checks.
* The “pipeline bias” concern was partially mitigated with controlled distillation and cross-family (Llama) experiments suggesting gains are not purely stylistic alignment.

**Concerns still outstanding:**

* The core tension around model-based correctness filtering remains. While ablations suggest reintroducing filtered items hurts performance, the procedure still risks skewing the dataset toward model-solvable, well-posed QA and away from genuinely challenging scientific reasoning.
* Methodological novelty remains limited, and it is still not fully clear what new, generalizable principle the community should take beyond careful engineering and filtering.
* The decontamination counts suggest substantial overlap removal for some benchmarks, and the paper would benefit from clearer interpretation of these numbers and concrete overlap examples to further de-risk leakage concerns.
* Full community value depends on complete release of dataset, models, and code, which is stated as “upon acceptance,” leaving residual uncertainty for reproducibility at decision time.

**Reviewer Scores:**

* Reviewer Qsbw: **4 → 4**. More details help, but novelty and conditional release remain, and key concerns are not fully removed.
* Reviewer Rf7n: **6 → 6**. Rebuttal strengthens analysis, but does not change the overall borderline stance.
* Reviewer 4dT2: **4 → 4**. Added decontamination and scaling details improve clarity, but filtering bias and limited novelty concerns largely persist.

---

### Decision · Program_Chairs · 2026-01-26

Reject